# High-resolution snapshots of human *N*-myristoyltransferase in action illuminate a mechanism promoting N-terminal Lys and Gly myristoylation

Cyril Dian [1,9], Inmaculada Pérez-Dorado[2,3,6,9], Frédéric Rivière[1], Thomas Asensio[1], Pierre Legrand [4], Markus Ritzefeld[2,7], Mengjie Shen[2,8], Ernesto Cota [3], Thierry Meinnel [1✉], Edward W. Tate [2,5✉] & Carmela Giglione[1✉]

The promising drug target *N*-myristoyltransferase (NMT) catalyses an essential protein modification thought to occur exclusively at N-terminal glycines (Gly). Here, we present high-resolution human NMT1 structures co-crystallised with reactive cognate lipid and peptide substrates, revealing high-resolution snapshots of the entire catalytic mechanism from the initial to final reaction states. Structural comparisons, together with biochemical analysis, provide unforeseen details about how NMT1 reaches a catalytically competent conformation in which the reactive groups are brought into close proximity to enable catalysis. We demonstrate that this mechanism further supports efficient and unprecedented myristoylation of an N-terminal lysine side chain, providing evidence that NMT acts both as N-terminal-lysine and glycine myristoyltransferase.

[1] Université Paris-Saclay, CEA, CNRS, Institute for Integrative Biology of the Cell (I2BC), Gif-sur-Yvette 91198, France. [2] Department of Chemistry, Imperial College, Molecular Sciences Research Hub, Wood Lane, London W12 0BZ, UK. [3] Department of Life Sciences, Imperial College London, Exhibition Road, South Kensington SW7 2AZ, UK. [4] Synchrotron SOLEIL, Gif-sur-Yvette, Cedex 91192, France. [5] The Francis Crick Institute, 1 Midland Rd, London NW 1AT, UK. [6] Present address: Evotec Ltd, 114 Innovation Dr, Milton Park, Milton, Abingdon OX14 4RZ, UK. [7] Present address: Evotec SE, Essener Bogen 7, Hamburg 22419, Germany. [8] Present address: Oakland Innovation, Mill Rd, Harston, Cambridge CB22 7GG, UK. [9] These authors contributed equally: Cyril Dian, Inmaculada Pérez-Dorado. ✉email: thierry.meinnel@i2bc.paris-saclay.fr; e.tate@imperial.ac.uk; carmela.giglione@i2bc.paris-saclay.fr

Glycylpeptide *N*-tetradecanoyltransferase 1, commonly known as *N*-myristoyltransferases (NMTs, EC 2.3.1.97), are universally conserved eukaryotic enzymes responsible for a major post-translational modification (PTM), myristoylation (MYR)[1]. NMTs are thought to transfer a C:14:0 acyl-CoA (MyrCoA) exclusively to the N-terminal glycine (Gly) of specific substrates. NMT-directed *N*-myristoylation is usually co-translational following initiator Met removal from nascent polypeptides by methionine aminopeptidases at the ribosome (Fig. 1a)[2]. MYR is fundamental to diverse cellular processes including recruitment and interaction of myristoylated proteins with membranes[3] and other proteins and can contribute to protein stability[4].

NMT was first characterised in yeast[5] and later in fungi, protozoa, insects, plants and mammals including humans[1,6]. Humans express two NMT isoenzymes (HsNMT1 and HsNMT2) in most tissues, accommodating hundreds of protein targets[7,8], including some implicated in diseases such as cancer and infection (for reviews see refs. [9–11]). NMTs are thought to be essential in all organisms and are validated drug targets, particularly for the treatment of infectious diseases[12–15]. However, progress in the field of human NMT biology has been undermined by non-selective and now invalidated compounds[16], raising questions about previous studies using these drugs. Further therapeutic development will be greatly enhanced by a complete understanding of NMT catalysis and substrate selectivity.

NMTs are members of the large GCN5-related *N*-acetyltransferase (GNAT) superfamily[17,18], which are involved in countless cellular functions and in a wide range of diseases and resistance mechanisms. GNATs catalyse donor acyl-coenzyme A (CoA) transfer to a variety of acceptor substrates[1]. Despite limited sequence homology, GNATs share a common 3D core architecture (Fig. 1b). Atomic-level mechanistic studies of GNATs have been hampered by a lack of structural data on GNAT proteins complexed with their natural ligands. Nonetheless, a common mechanism has been proposed, though definitive proof is still missing. For most GNATs, this Bi:Bi catalytic mechanism involves a general base which deprotonates the protein substrate amino group to generate a tetrahedral intermediate stabilised by a general acid. A proton wire of structured water molecules often connects the catalytic base to surrounding solvent[17,19], critically contributing to donor amine deprotonation and proton transfer from a base catalyst.

NMTs are atypical GNAT members featuring two adjacent GNAT domains with reduced sequence homology[20] (Fig. 1c, d). Previous studies have suggested that the MYR reaction does not resemble that of the other GNAT superfamily members[21]. The current accepted NMT mechanism, deduced from numerous NMT structures from different organisms in complex with MyrCoA or MyrCoA analogues and inhibitors[6,22], posits first the formation of a stable complex with MyrCoA in the N-terminal domain, which contributes to peptide acceptor site formation together with C-terminal GNAT domain residues[23] (Supplementary Fig. 1a–c). From observations arising from the structures of *C. albicans* (CaNMT) and *S. cerevisiae* NMT (ScNMT), MyrCoA binding-induced conformational switch of the conserved flexible 'Ab-loop' of NMT (Supplementary Fig. 1a–c), similar to

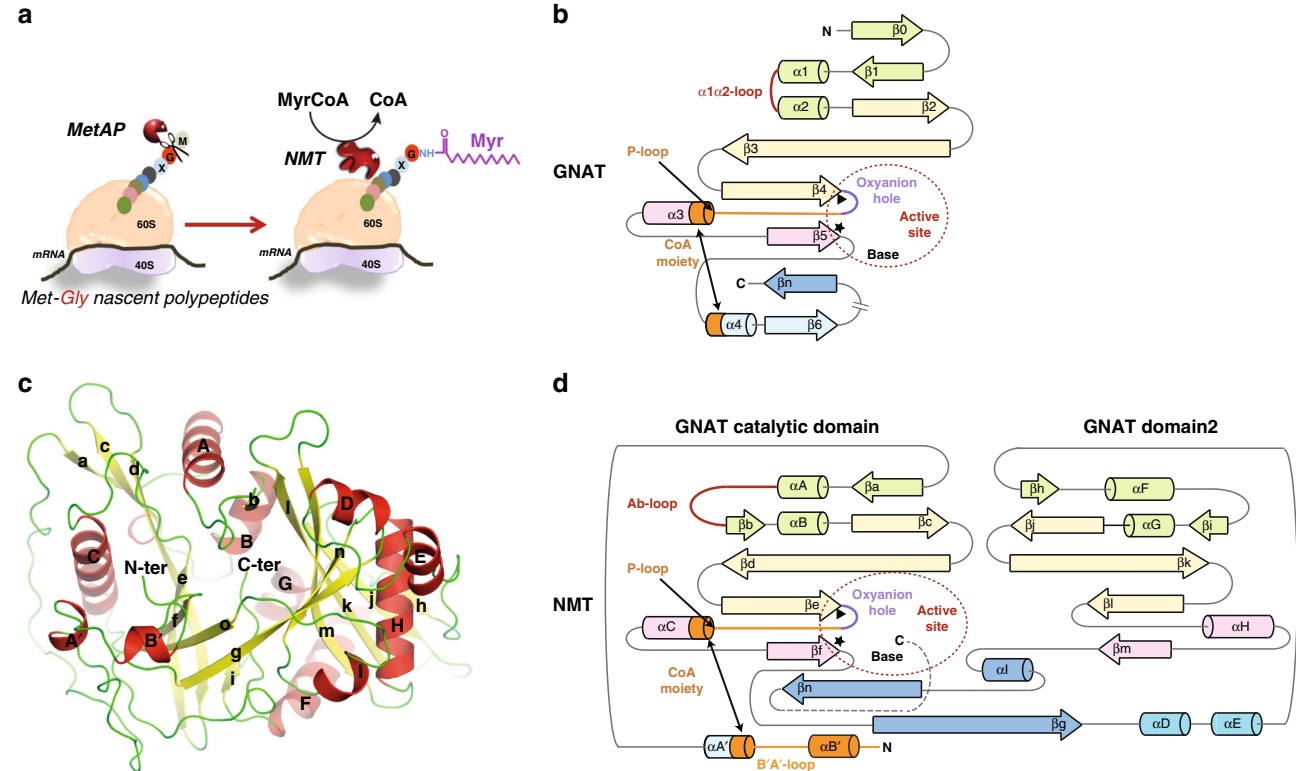

**Fig. 1 N-myristoylation reaction and organisation of GNAT and NMT. a** Cartoon representation showing the co-translational myristoylation process of the protein. The Met-Gly nascent polypeptides first undergo removal of initial Met by the action of methionine aminopeptidase (MetAP), and then the unmasked Gly is modified by NMT, which utilises MyrCoA to transfer the Myr moiety to the N-terminal Gly. **b** General topology of GNATs. Conserved secondary structural elements are coloured in pale green, yellow, pink and cyan, respectively. The non-conserved secondary structural elements are coloured in blue. Regions that are important for CoA binding are shown in orange. The oxyanion hole bulge is highlighted in violet. A red broken circle highlights the position of the active site. Ab-loop and equivalent α1α2-loop are coloured in red. Black stars and triangles highlight crucial positions in the active site. **c** View of HsNMT:MyrCoA fold (pdb = 5O9T) labelled with secondary structure elements according to sequence alignment of NMTs in Supplementary Fig. 2a. **d** NMT topology showing the GNAT-core fold using the same colour code as in **b**.

the α₁–α₂ loop of serotonin N-acetyltransferase[19] (Fig. 1b), is thought to participate in the formation of a competent peptide-binding site and contribute to overall enzyme efficiency, but a role in catalysis has yet to be defined[24,25]. However, involvement of the Ab-loop was controversial, as studies of full-length ScNMT suggested that the ordered-closed conformation of the Ab-loop, confining the peptide substrate within its binding site, was an artefact due to NMT missing the N-terminal region including the B′A′ loop[26].

The structure of *Saccharomyces cerevisiae* NMT (ScNMT; Supplementary Fig. 1d) in complex with a bound natural substrate peptide and a non-hydrolysable MyrCoA analogue (S-(2-oxo)-pentadecyl-CoA, NHM)[23] shows a salt bridge between the distal carboxy base and the primary amine of the Gly (Gly2) acceptor peptide, suggesting, differently from the common mechanism of other GNATs, that deprotonation of this amine is directly mediated by the base. However, in this non-functional ternary complex, the Gly2 amine is 6.3 Å away from the MyrCoA reactive thioester, meaning that the deprotonated Gly2 amine is predicted to rotate and move toward MyrCoA to allow nucleophilic attack of the thioester group[23]. This sole motion was suggested to be responsible of NMTs selectivity for Gly over Ala.

Here we report a series of high-resolution HsNMT1 structures co-crystallised with MyrCoA and reactive substrate peptides. These structures reveal snapshots across the entire catalytic mechanism including the reactive intermediate state, thereby providing the high-resolution map of MYR reaction chemistry. These structures highlight a number of important rearrangements of and around the amino-terminal acceptor group and provide an atomic-level deconstruction of each step of NMT reaction catalysis. Specifically, the reactive amino-terminal acceptor group undergoes highly dynamic interactions around a catalyst platform centred on Thr282, revealing a crucial role for a previously unseen water channel. We also show that the NMT long Ab-loop is a mobile structure that plays a crucial role in promoting pre-organisation of MyrCoA in the transition state, in turn triggering the concerted deprotonation of the substrate and nucleophilic attack of MyrCoA. Close examination of these structures provides valuable information about how to improve existing NMT inhibitors, particularly with respect to selectivity. Finally, our studies additionally support the transfer of a myristoyl moiety at the N-terminal epsilon amino group of Lys, decisively expanding the known NMT substrate range beyond the well-established N-terminal alpha moiety of Gly peptides.

## Results

### Overall structures of HsNMT1 binary and ternary complexes.
Previous structural studies have taken advantage of substrate or inhibitor analogues to investigate NMT catalysis. To capture different snapshots of NMT catalysis at high-resolution, we co-crystallised substrates with human NMT1 (HsNMT1) or soaked preformed crystals of HsNMT1/MyrCoA complexes with active peptide substrates featuring an N-terminal Gly residue (Gly2). We therefore solved the crystallographic structure of HsNMT1 in complex with two different peptides (X and Y; see Methods for peptide details) at 2.3 and 2.1 Å resolution, respectively. In addition, we recently reported the crystallographic complex at 2.7 Å resolution of HsNMT1 with a third peptide (Z)[8], which is further analysed here together in light of the X and Y complexes (see below). Each structure consisted of two HsNMT1 molecules in an asymmetric unit with the same overall fold (rmsd 0.157–0.236 Å). Further, this fold was conserved between HsNMT1 molecules of each solved complex (Supplementary Fig. 2) and between HsNMT1 and HsNMT2 structures previously reported in complex with MyrCoA or non-hydrolysable MyrCoA

NHM[2]. The compact GNAT core was clearly identified (Fig. 1c, d and Supplementary Fig. 2b) and consisted of an internal pseudo two-fold symmetry axis between the N-terminal half building the MyrCoA binding site, and the C-terminal region, shaping the peptide substrate-binding site. The latter comprised seven major pockets accommodating peptide substrate side chains of residues aa3 to aa6[8].

Our crystallographic structures revealed binary and ternary complexes in substrate, intermediate and product states trapped in crystallo along the *N*-myristoylation (MYR) reaction pathway: (1) peptide X and MyrCoA in a ternary complex with HsNMT1 prior to the reaction (HsNMT1:MyrCoA:X chain B, Fig. 2a, e); (2) the tetrahedral intermediate formed by the reaction of peptide Y N-terminal amine with MyrCoA thioester (HsNMT1:TI-Y chain A, Fig. 2b, f); (3) the myristoylated peptide and CoA products bound prior to release (HsNMT1:CoA:MyrX chain A and HsNMT1:CoA:MyrY chains A and B, Fig. 2c, g); and (4) the binary complex between HsNMT1 and the myristoylated peptide Z product (MyrZ) following ordered CoA release (HsNMT1: MyrZ, Fig. 2d, h[8]). The product complexes with peptides X and Y (HsNMT1:CoA:MyrX chain A and HsNMT1:CoA:MyrY chain B) fully overlapped, indicating that the state was independent of peptide composition (Fig. 3a). Of note, in chain A of the NMT complex with peptide Y, both tetrahedral intermediate and products of the reaction (CoA and MyrY) were observed, exhibiting occupancies of 0.3 and 0.7, respectively.

### Snapshots of HsNMT1-mediated MYR catalysis.
All three peptide substrates, despite their different sequences, adopted an extended β-strand conformation between aa2-aa7 (Fig. 3a), with the aa3-aa6 side chains located in the four side chain recognition pockets[8] featuring in strictly identical conformations before, during and after the MYR reaction (Fig. 3a, pockets 2–5).

HsNMT1:MyrCoA:X represents a snapshot of the peptide substrate prior to nucleophilic attack on MyrCoA (Fig. 2a, e). The free N-terminal Gly2 (aa2) of peptide X is hosted in a cavity (pocket 1, Fig. 3a) built by the side chains of Tyr180, Tyr192, Asn246, Thr282, Leu495, the C-terminal carboxylate of Gln496, and the myristoyl thioester of MyrCoA (Fig. 4a). The extended peptide conformation and Gly2 positioning orient the Gly2 peptide bond antiparallel to the MyrCoA thioester, with the amino-terminal group at ideal distance (3.4 Å) from the reactive MyrCoA thioester carbonyl (CM1, Figs. 3b and 4a). This carbonyl lies in the so-called oxyanion hole formed by the amide backbone of Phe247 and Leu248 (equivalent to Phe170 and Leu171 in ScNMT;[24] Fig. 4a). At the active site, Gly2 is constrained by three sets of H-bonds. The first set anchors Gly2 at the active site through direct interaction of its amino-terminal group with Thr282 (Fig. 4a). In the second, Asn246, Tyr180 and Tyr192 interact with the aa3 amide nitrogen via a water molecule (wat1) (Fig. 4a), ensuring correct substrate orientation toward the catalytic base C-terminal carboxylate of Gln496. The third set of H-bonds between Gly2 and the Gln496 carboxylate are mediated by a second water molecule, wat2 (Figs. 3b and 4a), at the end of a 22-Å-long, 544 Å³ solvent channel connecting the active site to solvent (Fig. 4 and Supplementary Fig. 3). These three H-bond networks contain strictly conserved residues (Supplementary Fig. 2a), suggesting that the water channel plays an active role in catalysis. These key mechanistic insights were missed in the previous ScNMT structure solved in complex with a peptide and NHM[23], since the latter was non-reactive. Superimposition of the structures reveals steric clashes in the ScNMT structure between Nα of Gly2 and the additional carbon atom of NHM that impede effective Gly2 positioning in pocket 1 (Fig. 3b), preventing both the peptide substrate and Ab-loop from adopting

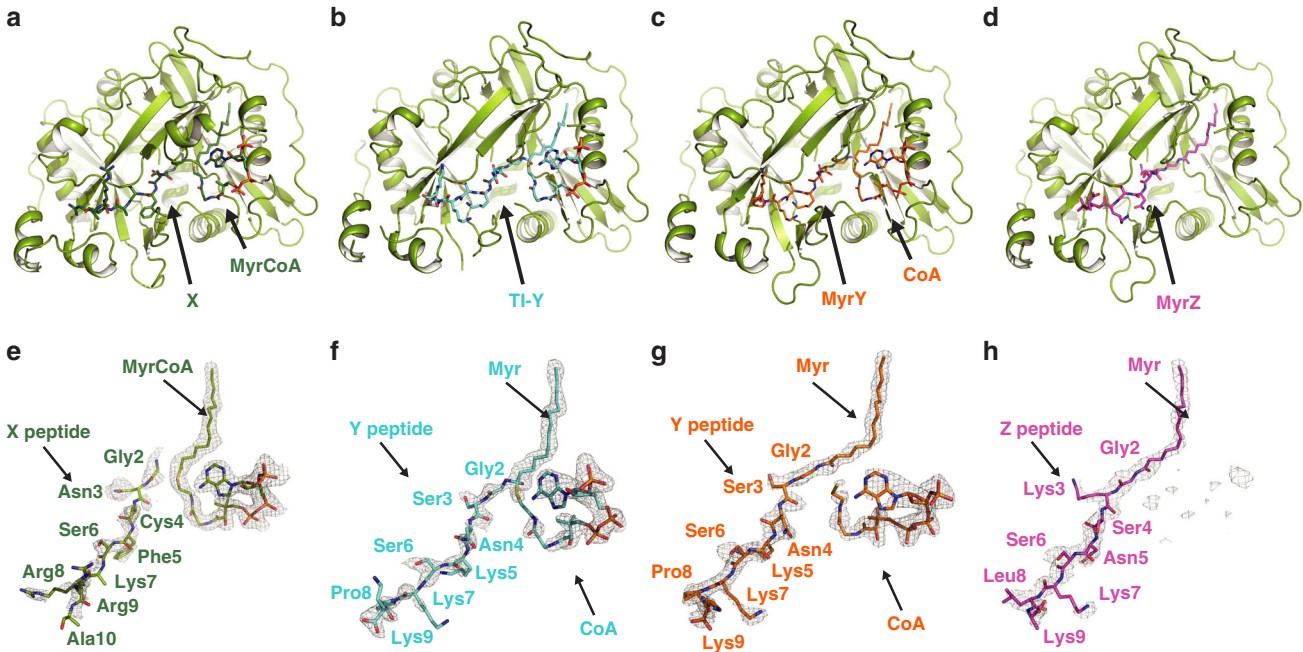

**Fig. 2 Overview of the obtained crystallographic NMT complexes.** Crystal structures of HsNMT1 in complex with substrates, TI and products obtained with different peptides X, Y and Z. The HsNMT1 main chain is displayed as a green ribbon. Substrates, TI and products are displayed as sticks and are indicated with black arrows. **a** Crystal structure of HsNMT1 in complex with MyrCoA and peptide X (green). **b** Crystal structure of HsNMT1 in complex with TI-Y, in which peptide Y and CoA are coloured in cyan. **c** Crystal structure of HsNMT1 in complex with MyrY and CoA (orange) products. **d** Crystal structure of HsNMT1 in complex with MyrZ product coloured in pink. **e–h** Detail of the "fofc omit" sigmaA-weighted electron density maps ($mF_{obs} − DF_{calc}$, $PHI_{calc}$), shown in grey, for substrates, TI and products (in sticks coloured as in **a–d**). The omit electron density maps were calculated after a round of refinement in which the occupancy of substrates, TI and products was set to zero. The e/Å$^3$ value of each "2fofc omit" sigmaA-weighted electron density map ($2mF_{obs} − DF_{calc}$, $PHI_{calc}$) at 0.92–1 rmsd level (0.25, 0.33, 0.33 and 0.30 e/Å$^3$ for X, TI-Y, Myr-Y and Myr-Z, respectively) was used as reference to set the contoured value of fofc omit electron density maps shown. Peptide, Myr, MyrCoA and CoA moieties are indicated with black arrows.

catalytically competent extended β—strand and closed conformations, respectively (Fig. 3b, c). These findings strongly suggest the ScNMT aa2-aa5 peptide conformation and Gly2-Gln496 carboxylate salt bridge are artefacts caused by the extra methylene of NHM[23].

To challenge the primary importance of the water molecule channel in mediating Gly2 deprotonation, we created two truncated NMT1 variants in which the last 2 or 3 amino acids were removed. The crystal structure of the HsNMT1 deletion variant with truncation of the last two amino acids, HsNMT1ΔC2, shows the peptide substrate orientated towards Glu244, which becomes the new catalytic base, prior to nucleophilic attack at the carbonyl group of the MyrCoA (Supplementary Fig. 4a, left). The 3D structure also reveals both an intermediate state in which the Gly2 is oriented towards the MyrCoA and a final state with the formed Myr-peptide (Supplementary Fig. 4a, middle and right). Interestingly, the structure of the complex prior to nucleophilic attack shows that Gly2 interacts with the new catalytic base Glu244 via two water molecules of the water channel (Supplementary Fig. 4a, left), confirming their importance for NMT catalysis. Accordingly, using an in vitro MYR assay (see Methods), we established that the truncation does not alter the protein turnover rate ($k_{cat}$) (Table 1) and observed a significant reduction in the enzyme's relative affinity to the peptide. This suggests that the C-terminus of the enzyme is also directly involved in correct peptide adjustment into the catalytic centre. This hypothesis was confirmed by the biochemical and structural characterisation of the HsNMT1 variant with further truncation of an additional amino acid (HsNMT1ΔC3). HsNMT1ΔC3 displays barely detectable enzyme activity (Table 1), and its 3D structure reveals

different conformations of the N-terminal region of the peptide substrate. This indicates that the low measured activity of HsNMT1ΔC3 is due to the high Gly2 mobility, most likely induced by the large space created by the C-terminal truncation (Supplementary Fig. 4b). The water network shaping the water channel of the NMT:MyrCoA:X complex is unmodified in the various crystal structures described later. Such ordered water molecules ideally bridge the reactive amino terminus next to a catalytic base, usually the carboxy terminus but also alternative carboxylic side chains such as in HsNMT1ΔC2. This indicates that tight ordering of the water molecules ensured by the water channel is crucial to the catalytic activity of NMT.

In the HsNMT1:TI-Y crystal structure, the tetrahedral intermediate results from nucleophilic attack by the Gly2 amine on the MyrCoA thioester (Fig. 2b, f). Comparison of HsNMT1: MyrCoA:X and HsNMT1:TI-Y reveals that the Gly2 amino-terminal group undergoes a 70° rotation around the Cα-C bond on forming the tetrahedral intermediate (Fig. 4b vs. 4a and Supplementary Fig. 5a), leaving a space filled by a new water molecule (wat3). Wat3 likely arises from the water channel as it extends the network. Wat3 stabilises the TI complex through a new H-bond network with the Ile245 carbonyl and Thr282 amide groups (Fig. 4b vs. 4a). Thr282 appears to play a pivotal role as a catalyst platform by inducing both concerted rotation and nucleophilic attack by the deprotonated Gly2 amine. Importantly, it also offsets the distortion induced by displacement of the newly formed tetrahedral thioester carbon towards the peptide Gly2 amine (Fig. 4b vs. 4a and Supplementary Fig. 5a). The observed tetrahedral intermediate geometry is in agreement with the stereochemistry of the reaction path (Fig. 4b vs. 4a, and Supplementary Fig. 5a; see also Bürgi et al.[27]).

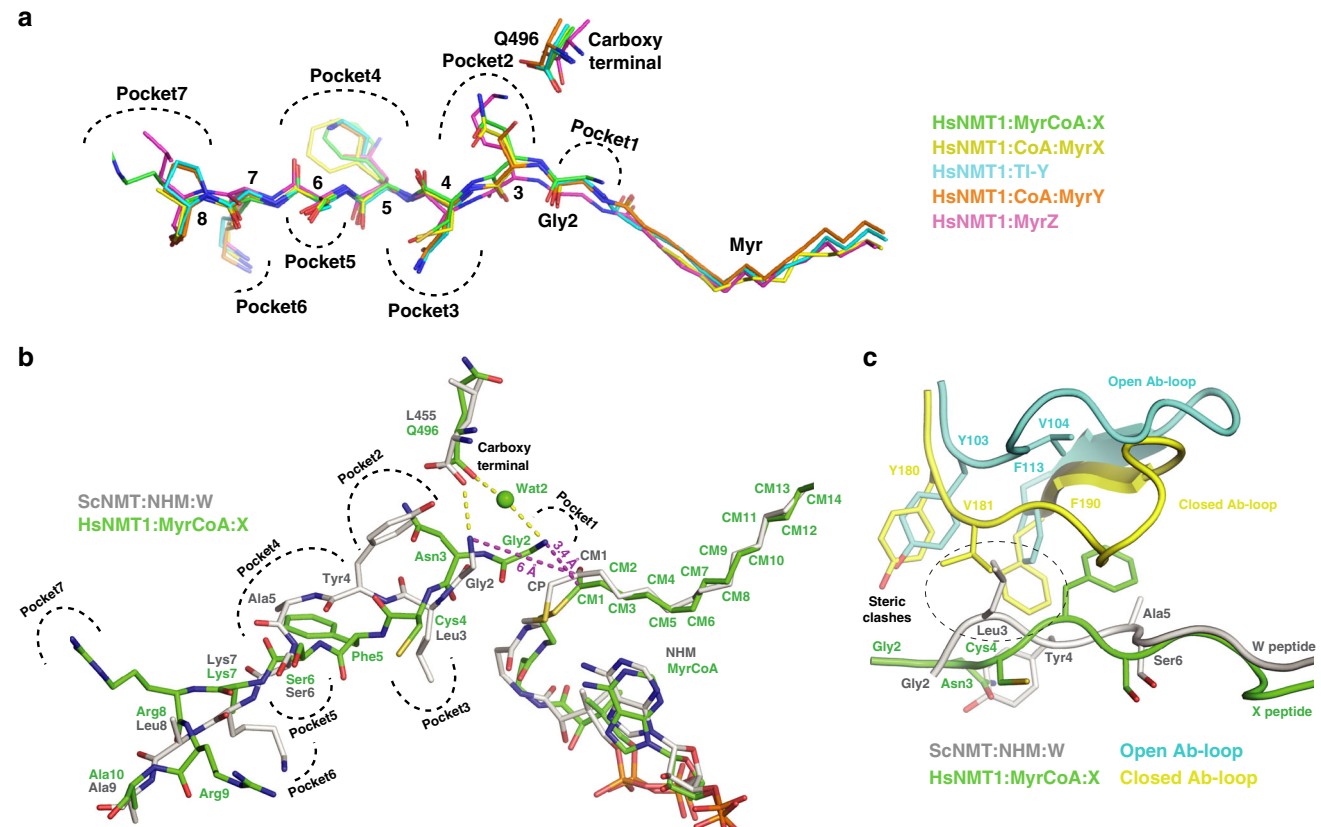

**Fig. 3 The β-strand conformation of the substrate peptide is appropriate for the Ab-loop closed conformation. a** Superimposition of the three peptides X, Y and Z in the different states reveals a conserved β-strand conformation. Gly2 from X peptide is bound into a new cavity, increasing to seven the number of "recognition pockets" of the peptide-binding groove. Pockets allocating each amino acid are indicated with black dashed lines. Peptides X (green), MyrX (yellow), TI-Y (cyan), MyrY (orange) and MyrZ (pink) are displayed as sticks. **b** Superimposed peptides from HsNMT1:MyrCoA:X (green) and ScNMT:NHM:W (grey, PDB:1IID). The 2-Å distance between Gly2 ammonium of peptide X and the extra-methylene (CP) of NHM shows how the analogue prevents Gly2 of peptide W to bind at pocket 1 and forces the Cα of Gly2 of W to shift back to occupy the n-1 aa position observed in peptide X. This motion influences the whole peptide W, including aa5, which co-locates with aa4 of peptide X in pocket 4 and illustrates the major consequences induced by the extra methylene group of NHM on peptide conformation, pocket occupancy and amino-terminal reactive group positioning. Magenta dashed lines indicate distances between each Gly2 ammonium and CM1 reactive carbonyl of MyrCoA. **c** Superimposition of HsNMT1:MyrCoA:X (green) and ScNMT:NHM:W (grey, PDB:1IID). The NHM-induced conformation of peptide W favours the Ab-loop open conformation (cyan) to the detriment of the closed conformation (yellow) observed in HsNMT1:MyrCoA:X due to steric clashes between Leu3 and Tyr4 from W peptide and V181 and F190 from HsNMT1:MyrCoA:X Ab-loop, respectively (black broken line circle).

The HsNMT1:TI-Y complex reveals key changes at the active site important for understanding both reaction progression (Fig. 4b vs. 4a) and substrate specificity. Tight packing of the Gly2 Cα against the Tyr180 hydroxyl, Asn246 carbonyl and MyrCoA thioester carbonyl groups in the TI structure (Fig. 4b) prevents accommodation of an Ala side chain (Fig. 4b and Supplementary Fig. 6), thereby contributing to specificity for Gly (Supplementary Fig. 6). Moreover, Tyr180 belongs to the conserved and mobile Ab-loop[26], which bears residues important for aa7 recognition, and provides a link between upstream substrate recognition and Gly2 selectivity.

HsNMT1-product structures reveal how CoA compacts following tetrahedral intermediate collapse, an event previously thought to be required to enable CoA release from NMT[23]. The β-strand conformation of the peptide induces a shift in the myristoyl carbonyl group toward the oxyanion hole (see asterisks in Supplementary Fig. 5b vs. 5a) and away from the sulphur atom of the CoA product (Supplementary Fig. 5b). The "question mark" shape of MyrCoA generates tension in the transition state that is released in the products, which causes the thiolate anion to move to 3.3 Å from the Myr-peptide amide bond (Supplementary Fig. 5b)[24] and adopt an optimal position to form an

intramolecular H-bond with the pendant amine of the CoA adenosyl ring (Fig. 4c and Supplementary Fig. 5). No acidic residues are sufficiently close to the thiolate group to permit its protonation; Tyr180 is the closest proton source at 4.0 Å from the thiolate leaving group. This suggests that TI reformation might occur via a 36° rotation around the C2P/C3P bond of CoA (Supplementary Figs. 5b and 4b), an observation that supports CoA release as a mechanism to prevent TI reformation as a key rate-limiting step of the catalytic cycle.

**The Ab-loop is a key structural element in the NMT catalysis.** The Ab-loop has been proposed to participate in the formation of an accessible peptide-binding site through a disordered-to-open transition induced by MyrCoA binding and is thought to contribute to overall NMT catalytic efficiency[23–25]. However, parallel studies on a ScNMT:MyrCoA:inhibitor complex[26] concluded that the closed Ab-loop conformation may be an artefact of N-terminal truncation in the construct used for crystallography (Supplementary Fig. 2a).

To address these uncertainties, we examined our structures to assess the effects of peptide binding and product formation on the

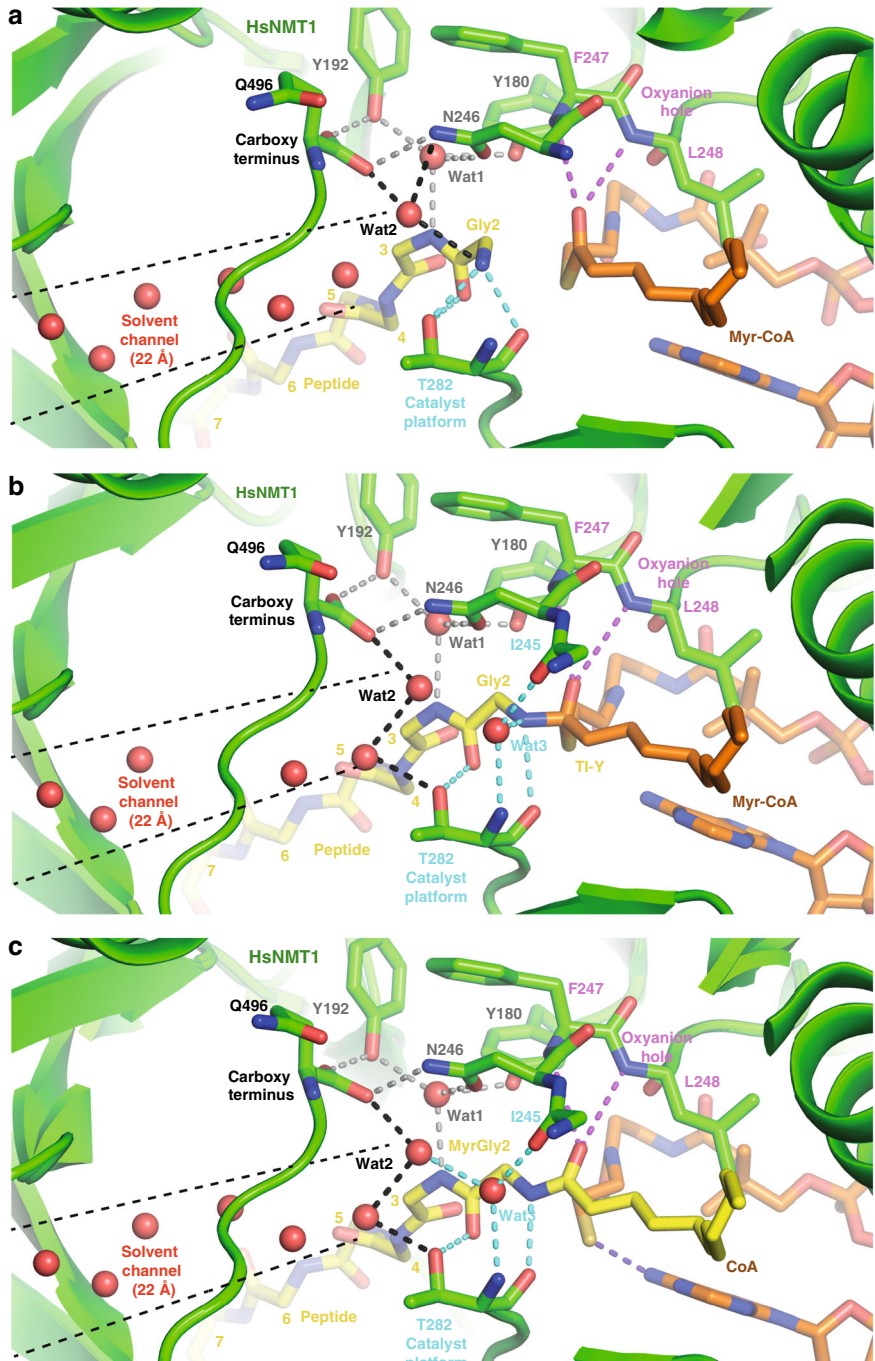

**Fig. 4 Three reaction snapshots reveal the importance of the T282 catalyst platform and water solvent channel.** View of the HsNMT1 active site showing direct and solvent-mediated hydrogen bonding interactions between the protein and: **a** substrates in HsNMT1:MyrCoA:X, **b** TI in HsNMT1:TI-Y and **c** products in HsNMT1:MyrX;CoA. The HsNMT1 chain is displayed as a green ribbon. Selected amino acids (green), substrate peptides (yellow), and MyrCoA and CoA moieties (both in orange) are shown as sticks. Hydrogen bond set1, linking Gly2 to the T282 catalyst platform, is shown as cyan dashes. Wat1-mediated hydrogen bond set2, linking Q496 to Tyr180/Tyr192 from the Ab-loop, substrate aa3 and Asn246 is shown as grey dashes. Wat1-mediated hydrogen bond set3 linking Gln496 to Gly2 ammonium is shown as black dashes. Water molecules from the solvent channel are shown as red spheres. Hydrogen bonds involved in the "oxyanion hole" are displayed as magenta dashed lines. The H-bond between thiolate and amine from CoA is shown as violet dashes.

Ab-loop conformation (Supplementary Fig. 7). This conformation was closed in all our structures, with the exception of the tetrahedral intermediate complex, in which it was partially disordered (Supplementary Fig. 7e). Our structures revealed six key structural Ab-loop features: (1) the salt bridge between substrate Lys7 and Asp471, which is the only specific interaction stabilising aa7 in the

presence of all peptides (Supplementary Fig. 7); (2) an additional interaction network stabilising the closed Ab-loop conformation in the HsNMT:CoA:MyrX and HsNMT:MyrCoA:X complexes (Fig. 5c vs. 5a, b); (3) the conserved Phe188 residue, which acts as a lid on selectivity pocket 4 in both Ab-loop open and closed conformations (Supplementary Fig. 8); (4) the steric interaction between Val181

**Table 1 Catalytic parameters for HsNMT1 and related mutants using MyrCoA.**

| HsNMT1 variant | $K_m$ (µM) | $k_{cat}$ (s$^{-1}$) | $k_{cat}/K_m$ (s$^{-1}$.M$^{-1}$) | Relative $k_{cat}/K_m$ |
|---|---|---|---|---|
| Wild-type | 18 ± 4 | 0.41 ± 0.01 | 23,502 ± 5259 | 100 |
| ΔC2 | 629 ± 127 | 0.34 ± 0.04 | 538 ± 122 | 2.3 |
| ΔC3 | 2140 ± 134 | 0.22 ± 0.01 | 93 ± 5 | 0.4 |
| Lys107Glu/Lys252Glu | 51 ± 15 | 1.33 ± 0.14 | 25,848 ± 7846 | 110 |
| Tyr180Phe/Asn246Ala | 33 ± 11 | 0.68 ± 0.07 | 20,924 ± 7398 | 89 |
| Tyr180Ala | 765 ± 204 | 0.16 ± 0.02 | 206 ± 63 | 0.9 |
| Tyr180Pro | nm | nm | <2 | <0.01 |
| Val181Ala | 20 ± 5 | 0.36 ± 0.03 | 18,541 ± 4950 | 78 |
| Val181Leu | 54 ± 12 | 0.33 ± 0.03 | 6172 ± 1516 | 26 |
| Tyr192Ala | 187 ± 34 | 0.93 ± 0.10 | 4960 ± 1056 | 21 |

The GlyCysSerValSerLysLysLys (SOS3) peptide was used as reference peptide.
Nm indicates that no reactivity was observed at the lowest measurable rate value performed with the highest NMT concentration (10 µM).

and the peptide aa3 carbonyl, limiting Ab-loop closure (Fig. 5b, c, magenta dotted lines); (5) a positively charged peptide residue at aa7, favouring interaction of Asp471 and the negatively charged Ab-loop in the closed conformation (Fig. 5c and Supplementary Fig. 7); and (6) distortion of the MyrCoA thioester bond when the Ab-loop is closed (Fig. 5c, d).

The charged peptide substrate aa7 (Lys) plays an important role in peptide selectivity and catalysis and is confined within an outer-inner interface cavity (pocket 6) (Fig. 3a). The back and front walls of this cavity are formed by hydrophobic residues surrounding the aliphatic chains of aa7 and Asp471, respectively (Supplementary Fig. 7a–c), whilst the cavity roof is built from three aspartate residues (Asp471 and Asp183/Asp185 of the Ab-loop), salt bridging the aa7 amine (Supplementary Fig. 7a, b). Although this cavity has evolved to accommodate a charged aa7, its side chain is disordered in HsNMT1:MyrCoA:X in contrast to HsNMT1:TI-Y, where the Ab-loop is disordered and the substrate aa7 side chain interacts with Asp471 (Supplementary Fig. 7d, e). These observations suggest that Asp471 initiates aa7 binding in the active cleft and that aa7 does not trigger the closed conformation of the Ab-loop but instead contributes to its transient stabilisation after the peptide is bound.

Comparing ScNMT:MyrCoA (open) and HsNMT1:MyrCoA:X (closed) (Fig. 5b, black circle), the closed switch of the Ab-loop forces MyrCoA to change conformation, shifting its C6 atom by 1.77 Å and distorting the thioester 20° out of sp$^2$ planarity (Fig. 5d). This observation is supported by reanalysis of reported structures of ternary complexes with inhibitors at the peptide binding site, which show that distortion of the MyrCoA thioester bond only occurs when the Ab-loop is closed (Supplementary Fig. 9). Together, these findings support a role for the Ab-loop in TI formation by pre-organising MyrCoA toward the TI state in a concerted manner with the oxyanion hole (Fig. 4a, pink dotted lines, right hand side), driving thioester bond polarisation when a peptide correctly fills the substrate-binding groove.

To verify the importance of the closed Ab-loop conformation in NMT catalysis, we generated a number of variants targeting key residues of this loop and we assayed their activity. Variants Val181Leu or Tyr192Ala showed reduced NMT catalysis (Table 1). Most interestingly, the variants Tyr180Ala and Tyr180Pro induced reduced and complete loss of MYR activity, respectively (Table 1). Indeed, Tyr180 is likely to be implicated in both MyrCoA conformational change (Fig. 5d) and Gly2 scanning (Supplementary Fig. 6) suggesting a strong link between activity and selectivity in NMTs. As a Tyr180Phe/Asn246Ala substitution did not alter catalytic efficiency with respect to the wild-type, we concluded that the hydrophobic character of residue 180 and its backbone were crucial for the Ab-loop to adopt the required structural constraint on MyrCoA. Tyr180 is positioned at the very

kink of the loop and locks one side of the Ab loop (Fig. 3c). Taken together, these biochemical and structural data highlight the importance of the transient Ab-loop closed conformation in NMT catalysis.

**Impact of the B′A′ N-terminal α-helix on product release.** The HsNMT:MyrCoA:X complex was obtained with a long NMT construct (see Methods) bearing the N-terminal B′A′-loop. Interestingly, the B′A′-loop is fully ordered and partially folds as an α–helix in this complex (Figs. 5b, c and 6a, and Supplementary Fig. 10). Such folding of the B′A′-loop further stabilises both MyrCoA/CoA via several interactions with the CoA moiety (Fig. 5b, c, bottom left and 6a). First, the P3 phosphate of MyrCoA/CoA makes three hydrogen bonds with the backbone nitrogen atoms of Gln118 and Tyr117 and the side chain of Arg115 (Fig. 5e). Second, the B′A′-loop makes interactions with the fg-loop (Fig. 5e, pink), which favour the proper planar orientation of the hydrophobic interactions of the adenine CoA ring provided by both Leu287 and Val285 side chains. The adenosine ring as a results stacks in between this pocket and the panthotenic-β-alanine side of CoA. The side chain of Val286 (part of fg-loop) is settled in a hydrophobic cavity made by the side chains of Ala196, Met109, Ala112 within the helical part of the B′A′-loop region. Stabilisation of the fg-loop, is achieved by direct hydrogen bonding between the Ala106 carbonyl (αB′ helix) and the Val286 backbone nitrogen. Third, Pro105 acts as a top lid to the ribose of the adenosine moiety. Finally, a Lys107-Asp184 salt bridge strengthens the B′A′-loop fold linking B′A′-loop with the closed conformation of Ab-loop (Fig. 5b, c). As a result, reinforced interactions of the B′A′-loop with the MyrCoA are expected to mainly participate in closing of the loop (Fig. 5e). A variant including a Lys107Ala substitution showed a significantly increased $k_{cat}$ value (Table 1), in keeping with CoA release as the limiting step of the reaction. A B′A′-loop can be identified in almost all NMTs (Supplementary Fig. 2a), and despite a low level of length and homology it is likely that similar interactions occur in other NMTs, as exemplified by the replacement of Lys107 and Glu184 by two hydrophobic residues in CaNMT.

The previously reported HsNMT2:MyrCoA complex (PDB code 4C2X) was solved with a construct partially lacking the B′A′-loop (Supplementary Figs. 2 and 10a). In this structure, the residual B′A′-loop is disordered despite the ordered conformation of the Ab-loop, suggesting that the complete B′A′-loop sequence is important for the proper folding of this region. The B′A′-loop conformation reverts to a disordered state in the product complex (HsNMT1:MyrZ) in which CoA has been released (Fig. 6b), driving opening of the distal part of the active site and unmasking the myristoyl group normally buried under

**Fig. 5 MyrCoA and Ab-loop relationship and its importance in Myr catalysis. a** Structural superimposition of Apo-CaNMT (PDB 1NMT, chain B, green) and the binary ScNMT:MyrCoA structure (PDB 1IIC, chain X, blue) showing rearrangements in the N-terminus (34–56, in cyan) and A-loop upon MyrCoA binding. The latter switches the Ab-loop conformation from disordered (Apo-CaNMT, in yellow) to open (in ScNMT:MyrCoA, in cyan) due to steric clashes (red dashed lines) with Glu109. Interactions with Arg178 and Lys181, stabilising the open Ab-loop conformation, are shown as black dashed lines. **b** Superimposition of ScNMT:MyrCoA (PDB 1IIC, blue) and HsNMT1:MyrCoA:X (green) complexes showing MyrCoA compaction induced by the Ab-loop closed conformation observed in the complex with peptide X (yellow), which is induced by steric clashes (red dashed lines) with Ab-loop Val181 (yellow). Detail of the folded B'A'-loop (yellow) induced by the closed conformation of the Ab-loop (yellow) is also shown, as well as its interactions with MyrCoA and Ab-loop (black dashed line). The Ab-loop closed conformation is limited by clashes with aa3 carbonyl from peptide X. **c** Superimposition of HsNMT1: MyrCoA:X (green) and HsNMT1:CoA:MyrX complexes reveals that the Ab-loop-induced compaction of MyrCoA (yellow) is also transmitted into the CoA product (cyan). Interactions stabilising the Ab-loop closed conformation in HsNMT1:CoA:MyrX by salt bridges are highlighted with black dashed lines. **d** Side-view zoom showing MyrCoA thioester planes and relevant Ab-loop residues as sticks, in both ScNMT:MyrCoA (cyan) and HsNMT1:MyrCoA:X (yellow), reveals an interbend 1–2 compaction of MyrCoA caused by the closed Ab-loop conformation in HsNMT1:MyrCoA:X. **e** View of the interactions of B'A'-loop region with MyrCoA moiety in HsNMT1:MyrCoA:X. HsNMT1 is displayed in cartoon, main core, B'A' region, Ab and fg -loops are colour in green, blue, yellow and pink respectively. Selected side chains and MyrCoA are displays as sticks. Hydrogen bonds and salt-bridges are shown as yellow dashes.

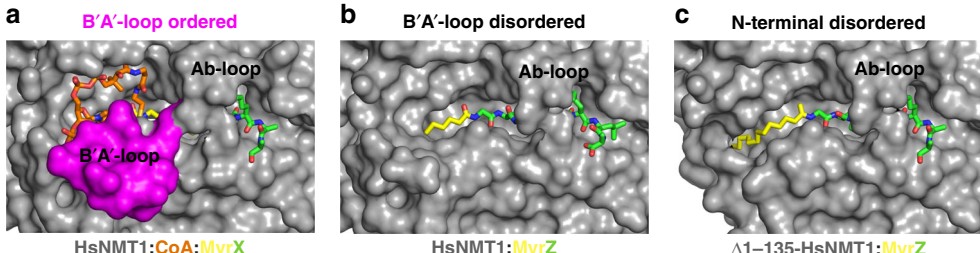

**Fig. 6 Role of the B'A'-loop in sequential release of the two products.** Solvent accessibility of the products at the HsNMT1 surface (dark grey) with the B'A'-loop surface highlighted in pink. Peptide, Myr and CoA moieties are shown as sticks in green, yellow and orange, respectively. **a** HsNMT1:MyrX:CoA crystal structure shows that an ordered B'A'-loop stabilises the CoA moiety and embeds Myr-peptide from the solvent. **b** HsNMT1:MyrZ crystal structure shows that CoA release, induced by unfolding the B'A'-loop, reveals a partially unmasked Myr moiety. **c** Model of HsNMT1:MyrZ deleted from the first 135 residues, mimicking the reported Apo-NMT crystal structure (pdb 1NMT), shows that the Myr moiety would be fully accessible to the solvent favouring Myr-peptide release or MyrCoA competition.

the CoA, facilitating Myr-peptide product release (Fig. 6b). Myr-peptide is a competitive inhibitor of MyrCoA binding[28], while CoA acts as a poor non-competitive inhibitor, stressing the importance of the B′A′-loop disordered conformation in Myr-binding site opening and product release[1,28]. These observations highlight a dual role for the B′A′-loop in stabilising the CoA and a closed Ab-loop (Fig. 6a) and in unmasking the myristoyl moiety to facilitate Myr-peptide product release.

We next considered how product formation influences B′A′-loop unfolding. Compaction of CoA in HsNMT1:CoA:MyrX (Fig. 5c, d) by the Ab-loop is further amplified by an intramolecular H-bond between the amine and thiol groups of NMT-bound CoA (Fig. 4c, violet dotted lines, in right hand-side). This contrasts with the elongated CoA conformation observed in acetyl-CoA (AcCoA) or CoA-bound structures from members of the GNAT family (Supplementary Fig. 11), in which the 3′-phosphoadenosine moiety is exposed to solvent with no specific interactions identified at the active site, thereby allowing diverse conformations (Supplementary Fig. 11). These observations are consistent with the hypothesis that relaxation of the tensed CoA in NMT could occur from the adenine ring, inducing N-terminal B′A′-loop disorder and CoA release from the binding site.

Taken together, these analyses establish: (i) a causal relationship between the ordering of the N-terminal B′A′-loop with the closed conformation of the Ab-loop when the HsNMT1 active site is loaded with a substrate peptide, and (ii) the disordering of the N-terminal B′A′-loop induced by CoA relaxation, which in turn releases first CoA and next Myr-peptide products.

**Reconstructing catalysis dynamics from crystal snapshots**. Interpolation of HsNMT1 ternary complexes suggests a putative trajectory for the NMT mechanism at high-resolution (Fig. 7) and identifies the conformational changes required for catalysis and apo-NMT recycling (Supplementary Movie 1). First, a nascent polypeptide chain with a positively charged Gly aa2 combined with an aa3-aa7 sequence compatible with pockets 2–6 enters and binds in the peptide-binding groove of HsNMT1:MyrCoA in an extended β-strand conformation (Fig. 7a). In this conformation, Gly2 is at an ideal distance and orientation from the carbonyl group of MyrCoA thioester, which is polarised in the oxyanion hole. A match between substrate and the substrate-binding groove switches the Ab-loop to a closed conformation (Figs. 4a, 5b and 7b), which translates into distortion of the MyrCoA thioester bond, leading to concerted water-mediated deprotonation of substrate Gly2 and nucleophilic attack on the thioester carbonyl guided by the Thr282 catalyst platform. The Ab-loop switch thus ensures tight and specific substrate selectivity and a β-strand conformation for the peptide (Fig. 7c). Reinforcement of the water-mediated interaction network around the peptide combined with thioester plane distortion enable NMT to lower the activation energy, while increasing its affinity for the tetrahedral transition state (Figs. 4b and 7d).

This detailed NMT mechanism is fully supported by existing site-directed mutagenesis quench-flow pre-steady state studies, which indicate that tetrahedral intermediate collapse is rate-limiting and that both ammonium deprotonation and nucleophile attack are ~140-fold faster than the steady-state rate[21], in agreement with Figs. 4b and 7c and the potential reversibility of Figs. 4b and 7d prior to release of CoA. Unlike previous structures, our data are also fully consistent with retention of activity in C-terminal truncations, since the water channel can bridge the missing residues[26,29] (Supplementary Fig. 4). Similarly, the decreased activity reported for ScNMT[Thr205Ala] and ScNMT[Asn169Leu] variants agrees with the importance of

analogous Thr282 and Asn248 in HsNMT1 (Supplementary Fig. 12) for the catalytic platform and Gly2 positioning, respectively. Furthermore, the importance of the N-terminal B′A′-loop in stabilising the ordered-closed Ab-loop is supported by a 10-fold reduction in peptide $K_m$ when this loop is deleted in ScNMT.

In the final steps of the mechanism, the new tensed CoA products and Myr-peptide are still bound to the active site (Fig. 7e), and subsequent relaxation of tensed CoA favours their sequential release (Fig. 7f). Spontaneous deprotonation of the Gln496 C-terminus may occur during this step; in the HsNMT1:MyrZ crystal structure, two rotamers of the Lys3 side chain suggest different protonation states for the C-terminal Gln496 carboxyl in each HsNMT1 molecule of the asymmetric unit. Finally, a closed-to-open Ab-loop switch and CoA-induced N-terminal NMT1 unfolding may induce Myr-peptide product release or alternatively allow competition by MyrCoA substrate to displace the Myr-peptide product.

**NMT catalyses MYR on the ε-NH₂ of Lys mimicking the α-NH2 of Gly**. NMTs are thought to exclusively acylate N-terminal Gly residues. Nevertheless, the ε-amino groups of internal Lys of several proteins including tumour necrosis factor-α[30], interleukin-α1 precursor[31] and SHMT2[32] have been shown to undergo MYR. Our structural data strongly suggest that both Gly- and Lys-MYR could be substrates for NMTs, a previously unprecedented class of substrates for this enzyme class. We recently reported the crystal structure of a complex of HsNMT1, MyrCoA, and a peptide sub-strate devoid of its amino-terminus i.e., the N-terminal Gly2 residue substituted with an N-acetyl group attached to an N-terminal Asn[8]. Compared to the complexes made with genuine peptide substrates, this 3D complex showed rotation around the Cα of aa3 causing the N-acetylated amide to enter the cavity normally occupied by the side-chain of residue 3[8]. As a result of this motion, the Asn side-chain also rotated around its own Cα backbone toward the position usually occupied by the Gly2 of genuine NMT substrates[8]. Models based on this crystal revealed that the Lys (the only natural amino acid with a side-chain featuring such a reactive amino group) at position 3 of the substrate peptide may position its ε-amino group for potential MYR (Fig. 8a, b), while being compatible with the constraints imposed by side chains of Tyr180 and Asn246 in charge of Gly2 selectivity.

A dozen NMT substrates featuring a Lys3 have been identified in humans including Arf6 and several calcium-binding proteins of the hippocalcin (HPCA) family[8]. We therefore investigated whether peptides featuring Lys3 could be NMT substrates through either or both Gly and Lys moieties. To challenge this hypothesis, we first performed MALDI-ToF-ToF analysis of the HsNMT1-catalysed products of two Gly-Lys starting peptides derived from Arf6 and HPCA. The MS/MS spectra unambiguously revealed only MYR-Gly and no Lys-MYR (Fig. 9a and Supplementary Fig. 13a, b), fully in keeping with crystallographic data of the complex between HsNMT1 and the HPCA peptide (see Supplementary Fig. 3 in Castrec et al.[8]).

We next challenged whether blocking the alpha amino group of a Gly-Lys starting peptide with an N-acetylation, by analogy with the crystal structure featuring Cα3 rotation, could render the ε-amino group of Lys3 reactive for MYR. MS analysis of the reaction in the presence of HsNMT1 revealed that Lys-MYR occurred (Fig. 9b and Supplementary Fig. 13c). We further assessed whether shortening the acetyl-Gly-Lys to acetyl-Lys would also permit NMT-driven ε-MYR. MS analysis of the product of HsNMT1 catalysis using an acetyl-Lys starting peptide clearly demonstrated conversion to N-terminal acetyl-Lys-MYR (Fig. 9c and Supplementary Fig. 13d).

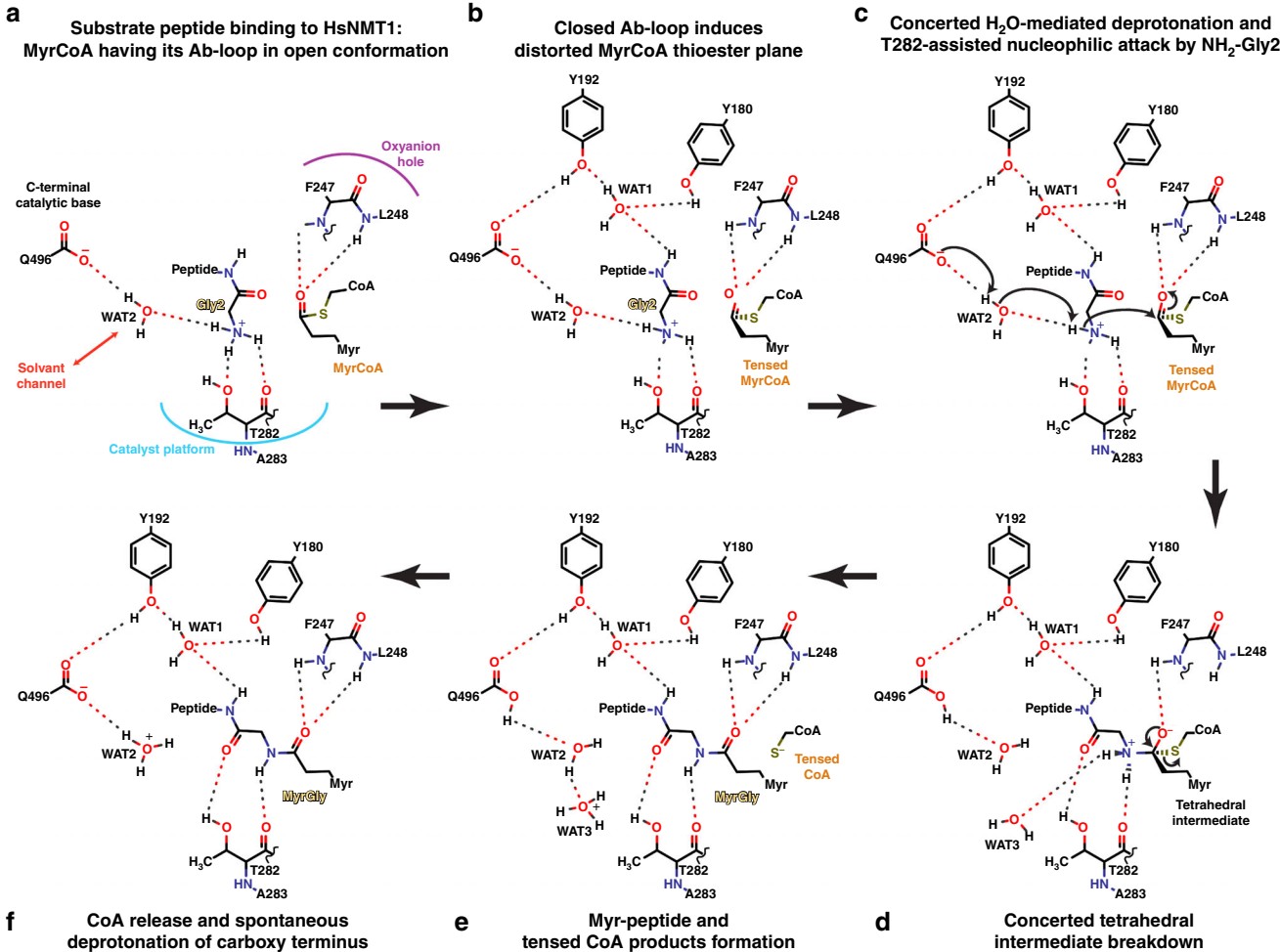

**Fig. 7 Detailed N-Myristoylation mechanism deduced from reaction snapshots. a** Substrate peptide binding to HsNMT1:MyrCoA having its Ab-loop in open conformation. **b** Closed Ab-loop induces distorted MyrCoA thioester plane. **c** Concerted H₂O-mediated deprotonation and T282-assisted nucleophilic attack by NH₂-Gly2. **d** Concerted Tetraedral Intermediate breakdown. **e** Myr-peptide and tensed CoA products formation. **f** CoA release and spontaneous deprotonation of carboxy terminus. Hydrogen bonds between wat3 and backbone atoms of I495 and T282 were not shown for clarity. The reaction dynamics are also detailed in Supplementary Movie 1.

To challenge whether only unavailability of the free amino group was responsible for Cα3 rotation leading to ε-MYR, we determined the crystal structure of HsNMT1 in complex with an acetyl-Gly-Asn starting peptide (Fig. 8c). As expected, there was similar rotation to the N-acetyl-Asn peptide. Occurrence of a free amino group by either Gly or Lys is crucial for proper orientation of both chains towards the pocket for aa3 or the active site (Fig. 8c). Finally, determination of the crystal structure of HsNMT1 in complex with an acetyl-Gly-Lys starting peptide clearly showed Lys-ε-MYR (Fig. 8d, e). Again, we could capture different steps of the ε-MYR reaction, prior and after nucleophilic attack, in the different monomers of a single crystal. We show that the Nε of Lys3 interacts directly with the C-terminus catalytic base, the Nε of Lys3 occupying the position of wat2 observed in the HsNMT1:MyrCoA:X complex (Fig. 4a vs 8d). This suggests that Lys-ε-MYR, in contrast to Gly-MYR, is not water-mediated and consequently highlights that the solvent channel is not necessary for Nε deprotonation.

## Discussion

The NMT catalytic mechanism described here provides an astonishing level of mechanistic sophistication involving motion in the N- and C-terminal domains together with mobile loops and provides a unified explanation for the importance of key conserved residues.

Examination of the structures of GNAT enzymes bound to different ligands or inhibitors suggests some common conserved features with NMT (Fig. 1a–c, and Supplementary Fig. 14a, b). The carbonyl group of the β5 C-terminal residue of some GNAT enzymes is likely to serve as a catalytic platform, placing the acceptor ammonium at an ideal distance and geometry relative to the thioester carbonyl. In the same vein, the last residue of GNAT β4 promotes the water molecule network (wat1/wat2/base) required for the ammonium group deprotonation, but only when AcCoA and acceptor are both bound. Identification of the water molecule network comprising wat1 and wat2 would likely help to track the position of the catalytic base in different GNAT superfamily enzymes. The GNAT α1α2-loop equivalent to the Ab-loop in NMT also seems to be required for acceptor ligand selectivity. Future determination of ternary complexes of other GNAT enzymes with both bound ligands will help to address the involvement of this loop in preforming the tetrahedral intermediate.

NMT is now well established as a valid target in parasitic and fungal diseases in which MYR is essential for the pathogen[33]. The

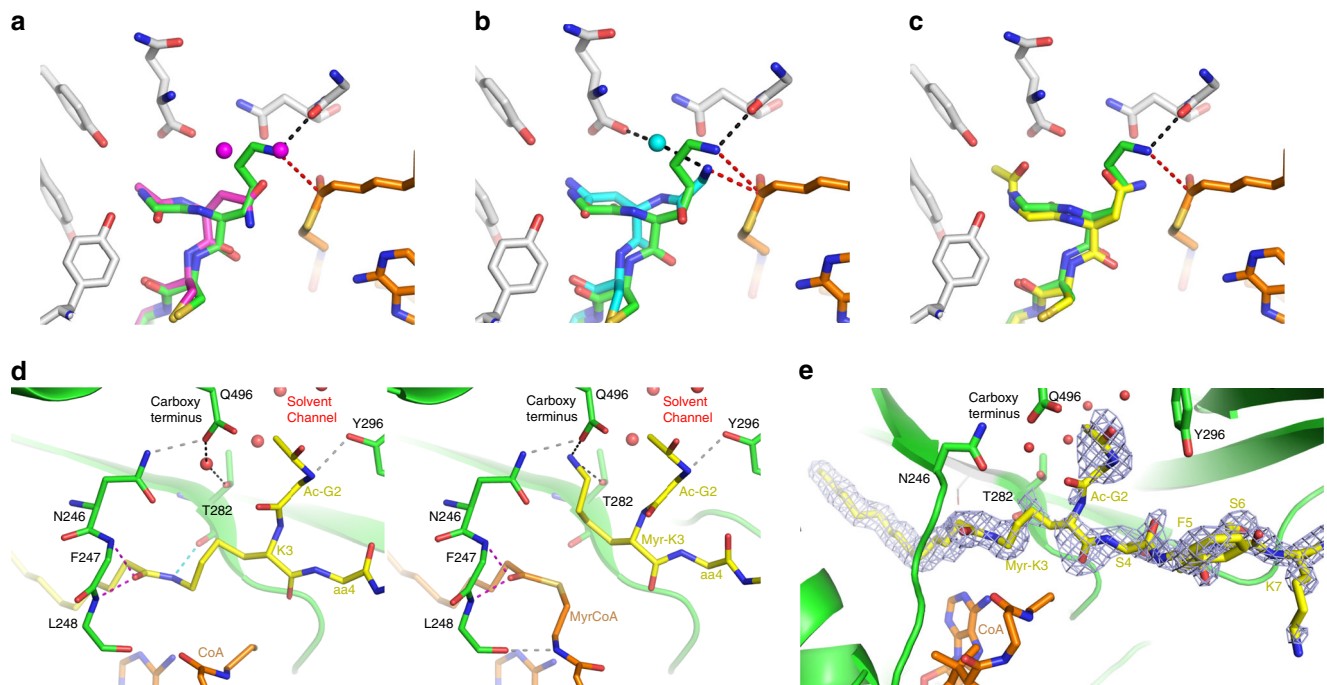

**Fig. 8 Structural analysis and characterisation of the binding of a Lys side chain into the active site reactive cavity.** The crystal structures 5O9T (HsNMT1/Ac-N₃CFSKPR), 6QRM (HsNMT1/ GN₃CFSKRRAA, this work) and 6SJZ (HsNMT1/Ac-GN₃CFSKPR, this work) were used in panels **a–c** to model the position of the side chain with the reactive epsilon amino group of a Lys in the active site instead of the alpha amino of a Gly2. **a** Gly-Lys modelled based on peptide Ac-Asn₃ of 5O9T (green) with 5O9T (peptide in magenta). **b** Same as **a**, but with Gly₂-Asn₃ from 6QRM (cyan). **c** Same as **a**, with peptide Ac Gly₂-Asn₃ (U) from 6SJZ (yellow). Selected amino acids of HsNMT1 are shown as grey sticks. MyrCoA moiety are shown as orange sticks. Wat1 linking the carboxy terminus of Gln496 to Gly2 ammonium is shown as a cyan sphere. Selected hydrogen bonds are displayed as black dashed line. Distance between either Nα (Gly₂) or Nε (Lys₃) and thioester carbonyl are displayed as red dashed line. **d** Snapshots of ε−Myr reaction trapped in crystal structure of HsNMT1 (green; 6SK2, V) in complex with MyrCoA (orange) and peptide V (Ac-GlyLysSerPheSerLysProArg; yellow) showing substrates (left; chain B) and reaction products (right; chain A). **e** Detail of the "fofc omit" sigmaA-weighted electron density map (mF_obs − DF_calc, PHI_calc), shown in blue, for Myr-V peptide in 6SK2 structure. The omit electron density maps were calculated after a round of refinement in which the occupancy of the Myr-V peptide was set to zero. The e/Å³ value of the "2fofc omit" sigmaA-weighted electron density map (2mF_obs − DF_calc, PHI_calc) at 1 rms (0.31 e/Å³) level was used as reference to set the contoured of the fofc omit electron density map shown.

dynamic catalytic mechanism revealed here opens up new opportunities for the design of inhibitors that exploit NMT plasticity. For example, there are now several examples of highly potent NMT inhibitors against both human and malaria parasite NMTs, but achieving a high level of selectivity has proven challenging to date due to high homology between NMTs at the substrate-binding site (Supplementary Fig. 14c–e). Indeed, this similarity was sufficient to develop IMP-1088, a 0.2 nM $K_d$ human NMT inhibitor with potent antiviral properties, starting from fragments originally identified as *Plasmodium* NMT inhibitors[33]. Alignment of the recently reported HsNMT1:IMP-1088 complex with the structures solved here highlights that the inhibitor occupies aa2-aa5 pockets, thus mimicking key aspects of peptide binding (Supplementary Fig. 14c–e). NMT amino acids involved in the interactions with both substrate and inhibitor molecules are conserved (Supplementary Fig. 2a), and a comparison of substrate-bound NMT and inhibitor-bound NMT suggests that more selective potency might be achieved through remodelling the interaction with the Gln496 C-terminal carboxyl group to maintain a water molecule at position +1 and repositioning a primary amine to react with MyrCoA, allowing the generation of a myristoylated inhibitor. Interestingly, examination of the closed Ab-loop intermediate structure reveals that the four residues forming the N-terminal helix starting from Pro105 in HsNMT1 or Leu20 in PfNMT of the B′A′-loop (see magenta box in Supplementary Fig. 14f) are not conserved in parasites (Supplementary Fig. 2a). No parasite NMT inhibitor design has

yet exploited this region of the binding pocket, and since this helix is critical for catalysis in HsNMT1 inhibitors, making specific interactions with the B′A′-loop of PfNMT may significantly increase selectivity. For instance, extension of the *N*-methyl moiety found in IMP-1088 could provide a means to probe this region of the pocket (Supplementary Fig. 14c–e).

We show that the uncovered NMT catalytic mechanism is fully compatible with ε−MYR of Lys residues located at the N-terminal side. When the N-terminus next to the Lys residue is blocked by an acylation event compatible with the size of the aa3 pocket, then Lys-MYR takes place. *N*-acetylation of Gly-Lys nascent peptides can be disfavoured since NatA – which may compete with NMT in vivo – generally selects negatively charged Gly-starting substrates[8]. Our data indicate that Gly-MYR is favoured over Lys-MYR when the alpha and epsilon amino groups are both available on the same peptide. The pK_a of the conjugate acid of the alpha amino group is below that of epsilon (7.7 ± 0.5 vs. 10.5 ± 1.1[34]) but close to both physiological pH (7.2 ± 0.2) and the pH of the assay performed at the optimal value for catalysis (pH 8.0; Towler et al.[5]). This might suggest that when both groups are available at the same N-terminal polypeptide, binding of the fully-protonated ε-amino group is favoured over the α-amino in the negatively charged environment of the aa3 pocket (see Fig. 1d in Castrec et al.[8]). This imbalance in favour of Gly-MYR could also result from the slightly longer side chain of Lys compared to Gly, which fits less well in the active site (Fig. 8d). Occurrence of Lys-MYR on an alpha amino blocked peptide indicates that the

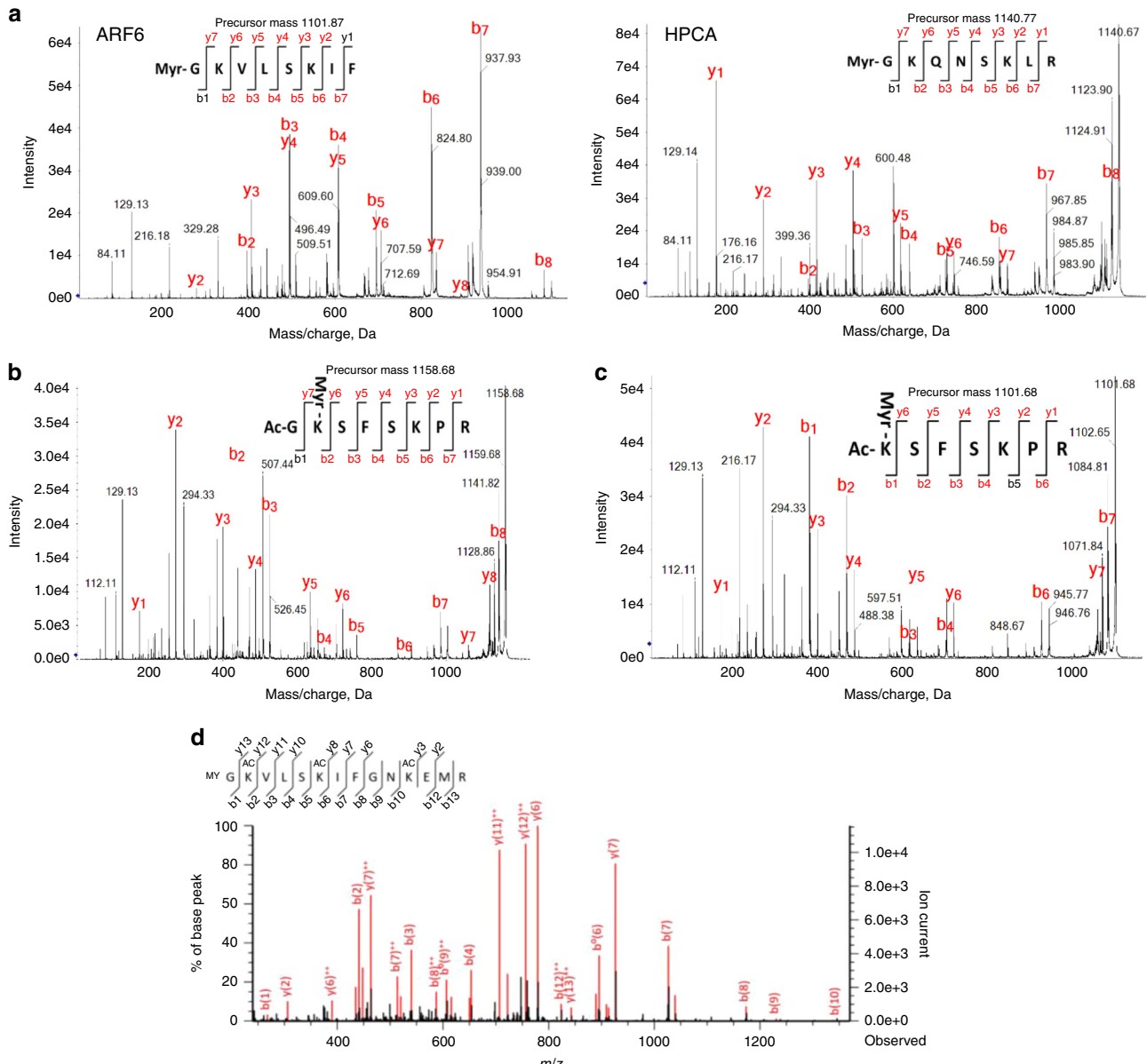

**Fig. 9 NMT catalyses Lys-MYR when the NH2 amino group is blocked. a–c** MS/MS spectra resulting from MALDI-ToF-ToF analysis performed as described in the Methods after NMT incubation with various octapeptides are shown. The control MS spectra in the absence or presence of NMT are displayed in Supplementary Fig. 13. MS/MS spectra from: **a** ARF6 (left) and HPCA-derived peptides (right); **b** AcGKSFSKPR (U peptide); and **c** AcKSFSKPR (V peptide). **d** Example of annotated MS/MS spectrum of the Myristoylated peptide GKVLSKIFGNKEMR from ARF6 (P62330); mass spectrometry of RT4 urinary bladder cells is as in Castrec et al.[8]. Acetylation of the Lys side chains of the protein sample was chemically induced in the presence of deuterated acetate. Both $y_{13}$ and $b_1$ ions favour Gly- over Lys-MYR.

modification is plausible in vivo on N-terminal Lys- or potentially Gly-Lys N-terminal peptides resulting, for instance, from post-translational proteolytic cleavage. However, we observed no such event in thousands of MS spectra produced in our labs over recent years or in a recent large-scale survey of human protein modifications[35]. This included Arf6, for which MYR was frequently observed in ESI-MS experiments after chemical acetylation of the sample to enhance detection of all free amino groups. MS/MS spectra indeed systematically favoured Gly- over Lys-MYR (Fig. 9d).

Why has no N-terminal Lys-MYR yet been identified in vivo? Two possible reasons include the likely reversibility of this modification and its putative role in an N-degron pathway. A number of active acyl hydrolases targeting Lys-MYR have been described over recent years including Sirtuin6, HDAC8 and HDAC11[32,36–38], making Lys-MYR a reversible modification dependent on the activity of these specific hydrolases, in contrast to the irreversible Gly-MYR. This implies that Lys-MYR, as with other reversible modifications (i.e., phosphorylation and ε-acetylation), might be intimately related to signal transduction pathways and tightly regulated by as yet unknown biotic or abiotic signals. Alternatively, it is tempting to speculate that N-terminal Lys-MYR may be part of an N-degron, and that an important role of Lys-MYR deacylases would be to ensure protein quality control by preventing intracellular Lys-MYR accumulation. Previous studies suggested an extension of the N-degron pathway[39] to Gly-starting proteins[8], which was supported by recent models in which conditional exposure of this degron

allows degradation of aberrant proteins that have escaped N-terminal MYR[4,40]. The Lys-MYR N-degron would further participate in this proofreading in a process dedicated to cellular MYR checking. This might be crucial at the physiological level given the number of major pathways in which known Gly-MYR proteins participate, preventing competition with irrelevant MYR proteins. Finally, our data do not support a mechanism or role for NMT in ε-MYR of internal Lys residues and further suggest that other GNAT members, including N-alpha acetyl transferases such as NatA or lysine aminotransferases, might also acetylate both Lys epsilon and alpha amino-terminal residues.

## Methods

**Protein production and crystallisation.** We used two different HsNMT1 constructs containing either 99–496 residues (Long-HsNMT1) or 109–496 residues (Short-HsNMT1). Long-HsNMT1 was used to solve the HsNMT1 complexes with peptides U, V, X and Z, while Short-HsNMT1 was used to solve the HsNMT1 complex with peptide Y. The open reading frames encoding two forms were both amplified by PCR using the full-length cDNA as a template. The PCR products were cloned into a modified pET28 vector featuring an N-terminal hexahistidine tag followed by Tobacco Etch Virus (TEV) cleavage site. Site-directed variants were obtained as described in Supplementary Methods. All primers used in this study are reported in Supplementary Table 3.

The recombinant plasmid encoding short-HsNMT1 was introduced into *Escherichia coli* BL21(DE3) pRareS. Cells were grown for 20 h at 18 °C in Overnight Express Instant TB auto-induction medium (Novagen). Cells were sonicated in 20 mM Tris-HCl (pH 7.5), 0.5 M NaCl (buffer A) supplemented with 10 mM imidazole, 1 mM MgCl$_2$, protease inhibitors, bovine pancreatic DNase I and Triton X-100 (0.1% (v/v)). The lysate was loaded directly onto a HisTrap FF Crude column (GE Healthcare, 1 mL) and protein eluted by a linear gradient of 10–500 mM imidazole in buffer A. The NMT pool was diluted 20-fold in 20 mM Tris–HCl (pH 8.9), 1 mM DTT (buffer B) before loading onto an anion-exchange column (GE Healthcare, 1 mL, HiTrap Q HP) and elution by a linear gradient of 25–500 mM NaCl in buffer B. Gel filtration on a Superdex 75 16/60 column (GE Healthcare) in 250 mM NaCl, 1 mM EGTA and 1 mM EDTA, 20 mM Tris-HCl (pH 8.5) buffer was used as final purification step. Short-HsNMT1 purified samples for crystallisation were concentrated to 20 mg/ml in 10 mM Tris–HCl (pH 8.5), 125 mM NaCl, 1 mM EGTA and 1 mM EDTA.

The long-HsNMT1 plasmid was introduced into *E. coli* BL21-PLysS Rosetta2 cells (Novagen). After growth at 22 °C, cells were lysed by sonication at 4 °C in 20 mM Tris–HCl (pH 8.0), 0.5 M NaCl, 5 mM 2-mercaptoethanol and 5 mM imidazole (buffer C). The lysate was loaded onto an immobilised nickel ion affinity chromatography (HisTrap Crude FF, 5 mL, GE Healthcare) and washed with buffer C with 25 mM imidazole. Elution was carried out at a 2-mL/min with buffer C plus 0.5 M imidazole with a linear gradient. The pool of purified protein was next dialysed overnight against 20 mM Tris–HCl (pH 8.9), 50 mM NaCl, 25 mM imidazole and 5 mM 2-mercaptoethanol in the presence of TEV protease (1 mg per 15 mg purified protein). His-tag free HsNMT1 was purified with a HisTrap Crude FF. The flow-through with HsNMT1 was diluted five times in buffer B and applied to an anion exchange chromatography column (HiTrap Q FF, 5 mL, GE Healthcare). HsNMT1 was eluted with buffer B supplemented with 0.2 M NaCl. Highly purified long-HsNMT1 fractions were pooled, concentrated to 15–30 mg/mL with an Amicon centrifugal filtration device (Merck Millipore) in 20 mM Tris–HCl (pH 8.0), 0.2 M NaCl, 1 mM DTT and stored at −80 °C before crystallisation trials. Protein samples for biochemical analysis were stored at −20 °C in the same buffer plus 55% glycerol.

Short-HsNMT1 was crystallised in complex with MyrCoA (HsNMT1:MyrCoA crystals) using the sitting-drop vapour diffusion method at 20 °C[41]. Crystals of HsNMT1:MyrCoA in complex with the N-terminal substrate peptide of the tyrosine-protein kinase Src (p60-Src) protein (peptide Y with sequence "Gly-Ser-Asn-Lys-Ser-Lys-Pro-Lys") were obtained by soaking HsNMT1:MyrCoA crystals in a cryosolution containing the precipitant components (16–18% (v/w) PEG 4K, 5 mM NiCl$_2$, 100 mM sodium citrate (pH 4.5) and 2.5% (v/v) glycerol), peptide Y at 150 mM and 25% (v/v) glycerol for 15 min. HsNMT1:MyrCoA:Y crystals were then flashed cooled in liquid nitrogen and stored for data collection. A complete X-ray data set at 2.10 Å resolution was collected at 100 K using synchrotron radiation at Diamond Light Source (Oxford, UK).

Crystals of long-HsNMT1 (HsNMT1:MyrCoA:X, HsNMT1:MyrCoA:U and HsNMT1:MyrCoA:V) were obtained by co-crystallisation using the hanging-drop vapour diffusion method at 20 °C. Crystallisation droplets were formed by mixing 2 μL of the of HsNMT1:MyrCoA:peptide complex (ratio 1:1.5:1.5) at 7.5 mg/mL (i.e., ~160/250/250 μM) with 2 μL of the precipitant solution containing 0.1 M MgCl$_2$, 0.2 M NaCl, 0.1 M sodium citrate pH 5.6 and 18–24% (w/v) PEG 6 K or 8 K. Crystals were cryoprotected in the reservoir solution supplemented with 15% (v/v) glycerol and flash cooled in liquid nitrogen. Complete X-ray datasets of HsNMT1:MyrCoA:X and HsNMT1:MyrCoA:U were collected at λ = 0.96770 Å and λ = 0.96600 Å at 100 K l in the ID30a3 and ID30a1 beamlines, respectively, at the European Synchrotron Radiation Facility (ESRF, France). Complete X-ray

dataset of HsNMT1:MyrCoA:V was collected at λ = 0.98400 Å from a single crystal at 100 K in the Proxima1 beamline, at the French National Synchrotron Facility (SOLEIL).

Datasets were integrated with XDS[42] and scaled and reduced using AIMLESS from the CCP4 package[43]. Crystals belonged to the space group P2$_1$2$_1$2 with similar unit cell parameters (Supplementary Table 1) and contained two NMT molecules per asymmetric unit. Structure resolution was accomplished in all cases using the molecular replacement method. The HsNMT1:MyrCoA:Y structure was solved using PHASER[44] and the HsNMT1:MyrCoA binary complex (PDB entry 4C2Y) as a search model. The structure of the HsNMT1:MyrCoA:X complex was solved using MOLREP[45] and protein coordinates of a Long-HsNMT1 model (PDB entry 5O9V) as a search model. The HsNMT1:MyrCoA:V structure was solved using PHASER[44] with PDB entry 5O9V as a search model. Structures were subjected to alternating refinement cycles using REFMAC-5 and PHENIX and manual model building using COOT[46–48]. NCS restrictions were applied. In both structures, the electron density maps allowed us to model most of both HsNMT1 molecules of the asymmetric unit, with the exception of the first 6 residues (residues 109–114) and some flexible loops (A182-184, A316-317, A408-413 and B313-318) in Short-HsNMT1:MyrCoA; and the first 6 residues (residues 99–104) in HsNMT1:MyrCoA:X, HsNMT1:MyrCoA:U and HsNMT1:MyrCoA:V, which were flexible and thus could not be traced. The good quality of the electron density maps also enabled the refinement of substrate peptide, reaction intermediate, and reaction product molecules bound to HsNMT1 in each complex. In the HsNMT1:MyrCoA:Y complex, linkers and cif files for the refinement of the MyrCoA-Y and the MyrY molecules were generated with JLIGAND as part of the CCP4i package[49]. In the refinement of the HsNMT1:MyrCoA:X, HsNMT1:MyrCoA:U and HsNMT1:MyrCoA:V complexes, chemical compound libraries were generated using the PRODRG server[50] in combination with eLBOW from the PHENIX suite. Difference electron density maps were calculated in PHENIX. The geometry of the final models was validated using MOLPROBITY[51]. Figures were generated using PYMOL (DeLano Scientific LLC, http://pymol.sourceforge.net/). X-ray data collection and refinement statistics are summarised in Supplementary Table 1 and 2.

**Fluorescence-based measurement of NMT activity.** NMT activity was assayed at 30 °C in a coupled assay as previously described[8]. Briefly, a reaction mixture containing MyrCoA and different concentrations of peptide acceptors was pre-incubated for 3 min at 30 °C before starting the reaction with HsNMT1. MYR kinetics were followed continuously for 10 min in triplicate, and the data were fitted over the first 2 min to obtain the initial velocity. The kinetic parameters ($k_{cat}$ and $K_m$) were obtained with the Enzyme Kinetics module 1.2 of Sigma Plot (version 9.0) by nonlinear Michaelis–Menten equation fitting.

**Mass spectrometry.** For MALDI-ToF/ToF analysis, 300 μL of a mixture containing 50 mM Tris (pH 8), 0.193 mM EGTA, 1 mM MgCl$_2$, 1 mM DTT, 5 μM sodium cholate, 0.04 mM Myr-CoA solution (stock solution 0.2 mM Myr-CoA, 10 mM sodium acetate, 2.5 μM sodium cholate), 0.5 μM NMT and 100 μM of synthetic peptide (Genscript, Piscataway, NJ) were incubated at 30 °C. The MYR reaction was followed over time by collection of 10 μL samples further diluted in 90 μL of water/acetonitrile solution. The different samples were then diluted five times in the matrix solution made of 5 mg/mL of α-cyano-4-hydroxycinnamic acid solubilized in water/formic acid/acetonitrile (50/50/0.1%). In all, 1 μL of each dilution was spotted on a metal target and dried. MS and MS/MS spectra of each sample were acquired with an AB SCIEX 5800 instrument in positive ion mode.

**Reporting summary.** Further information on research design is available in the Nature Research Reporting Summary linked to this article.

## Data availability

The 3D structures of NMT in complex with MyrCoA and the peptides reported here have been deposited at the PDB under codes 6QRM (peptide X), 6EHJ (peptide Y), 5O9S (peptide Z), 6SJZ (peptide U), 6SK2 (peptide V), 6SKJ (peptide S), 6SK8 (peptide T) and 6SK3 (peptide S). Other data are available from the corresponding authors upon reasonable request.

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

## Acknowledgements

This work was supported by MMV (grant 15–0054 to EWT), Cancer Research UK (Programme Foundation Award C29637/A20183 to EWT), the 7th Framework Programme of the European Union (PIEF-GA-2013-623648 to MR), and ANR PalMyProt grant to TM (ANR-2010-BLAN-1611-01). This work has benefitted from the support of the Labex Saclay Plant Sciences-SPS (ANR-10-LABX-0040-SPS), from the I2BC crystallisation platform supported by FRISBI ANR-10-INSB-05-01 and from the facilities and expertise of the I2BC proteomic platform SICaPS, supported by IBiSA, Ile de France Region, Plan Cancer, CNRS and Paris-Sud University. FR is supported by the PhD fellowship ARDoc program from Region Ile-de-France (17012695). We thank the staff at the I03 beamline, Diamond Light Source (Harwell Science and Innovation Campus, Oxfordshire, UK); the staff at the X-ray Crystallography Facility, Centre for Structural Biology at Imperial College London; Soleil for provision of synchrotron radiation facilities and the staff of Proxima1 beamline; the European Synchrotron Radiation Facility (ESRF) staff, particularly Matthew Bowler and Christoph Mueller-Dieckmann, for help with data collection; Remi Serwa for sequence analysis; Andy Bell for discussion and input on inhibitor development based on the structures here reported; and Yvain Nicolet for fruitful discussion concerning the NMT1 catalysis mechanism.

## Author contributions

C.G., E.W.T. and T.M. conceived the project and supervised the experiments. C.D. performed cloning, purification and structural analysis of 509S, 6QRM, 6SJZ, 6SK2, 6SKJ, 6SK8 and 6SK3. I.P.D. carried out the purification of 6EHJ and structural character-isation was performed by I.P.D. and C.D.; F.R. participated in cloning and protein purification and performed the MS experiments. T.A. cloned, purified all HsNMT mutants and participated in HsNMT1 biochemical studies and crystallisation of the majority of complexes. P.L. participated in structural characterisation of HsNMT1ΔC2 and HsNMT1ΔC3 complexes. M.R. synthesised and purified the Src N-terminal peptide used in the study. M.S. assisted in Short HsNMT1 protein production and crystallisation. E.C. provided support in structural characterisation and analysis of 6EHJ. C.G., C.D., T. M., E.W.T. and I.P.D. analysed the data; C.G., T.M. and C.D. wrote the manuscript with involvement of E.W.T. and I.P.D.

## Competing interests

The authors declare no competing interests.
