## [Peer Review File · Nature Communications]

Reviewers' comments:

Reviewer #1 (Remarks to the Author):

N-myristoyltransferase 1 (NMT1) is a key cellular enzyme which carries out lipid modification by facilitating the attachment of myristate to the N-terminal glycine of several protein molecules. Although structural works of NMT1 have been extensively studied, atomic-level mechanistic studies of NMT1 is not fully understood due to lack of structural data on NMT1 in complex with their natural ligands. Dian et al. firstly reported high-resolution human NMT1 structures co-crystallized with MyrCoA and reactive substrate peptides, showing substrate bound state, tetrahedral intermediate state and product state. Based on these structural studies, authors revealed a series of enzymatic mechanism of NMT1. They also revealed a novel mechanism of transferring myristoyl moiety at Lys in combination with structural and MS-MS analyses. These extensive studies would provide key missing links in our understanding of NMT1 activation and bring us one step closer to establishing a detailed and rigorous understanding of the NMT1 mechanism. Authors should address several concerns.

1) The authors represent 2Fo-Fc map of each substrate, TI and product. However, Fo-Fc map omitting with each substrate, TI and product is better to certify the model quality.

2) The authors crystallized NMT1 with peptide X or peptide Y in the presence of MyrCoA. The peptide X resulted in the crystals of NMT1:MyrCoA:X (substrate) and CoA:MyrX (product) while the peptide Y resulted in the crystals of NMT1:TI-Y (intermediate) and CoA:MyrY (product). Why did the authors succeed in obtaining the intermediate complex with peptide Y, but not with peptide X? Since the intermediate state is generally thought to be unstable, the precise information how to obtain the crystal under intermediate state is useful for general readers.

3) Page7, the author described that 'we therefore solved the crystallographic structure of HsNMT1 in complex with three different peptides~ at 2.4, 2.2, and 2.7 Å resolution, respectively'. The structures complexed with peptide X and Y are new structures while the structure with peptide Z (PDB code: 5O9S described in Data availability section (page 29)) was already reported in the other paper (Castrec, B et al., 2018 Nat. Chem. Biol. 14, 671-679.). This description is very confusing. The author must cite appropriately whenever they describe known results.

4) Page8 line 4 from the bottom, 'The free N-terminal Gly2 (aa2) of peptide X is hosted in a cavity (pocket 1, Fig. 3A) built by the side chains of Tyr180, Tyr192, Asn246, Thr282, Leu495; the C-terminal carboxylate~'
Leu495 should be replaced with Gln496 because the C-terminal residues is Gln496.

5) At page 15, the author mentioned that 'further stabilizing both MyrCoA/CoA via interactions with the ribose moiety~'.

Is there any specific interaction between the B'A'-loop and the MyrCoA/CoA?

Is there any specific interaction mediated by the B'A'-helix, which stabilizes the helix conformation of B'A'-helix? The interaction between Lys107 of the B'A' loop and Asp184 of Ab-loop seems to be the only interaction mediated by B'A' loop. However, Lys107 is not located at B'A'-helix, but at a loop region. It is not clear to me how the B'A' Nterminal α -helix is stabilized. In addition, the residues in the B'A'-loop including Lys107 is not conserved among NMTs. Does this less conservation imply the importance of the B'A'-loop is limited to human NMT1?

6) Page20, Fig. 13D and 13E seems to be typos (Fig S13D and S13E).

The sequence of peptide U seems not to be defined. Please define the sequence (probably Ac Gly2-Asn3?).

7) P35, at the legend of Figure5, interacting with Arg178 and Lys181 is typos? (Lys175 and Arg178?)

Figure 5A, "disorsered" would be "disordered".

8) Figure 2 E-H and Figure 3A-C, the peptides should be displayed in the same color. It is difficult to understand.

9) Figure 3, is the term 'pocket' generally used in NMT? Each pocket accommodate specific residue of the peptide for the recognition?

10) Figure 5C, it is quite difficult to identify the labels of residues in the Ab-loop because they are too close to each other (especially D183 and D185). Please place the labels clearly to identify them easily.

Reviewer #2 (Remarks to the Author):

The study by Dian et al. presents the first crystal structures of human N-myristoyltransferase 1 (NMT) in complex with its native substrates. NMT is a key cellular enzyme that transfers myristate to the N-terminal glycine of many eukaryotic proteins and is attractive therapeutic target for treatment of HIV infection and cancer. This study reveals new details of the interactions between protein residues and ligands at the active site at each step of the catalysis that provide inclusive depiction of the kinetic mechanism. The description of NMT complex structures is supported by mutagenesis experiments, fluorescence-based kinetic assays and mass spectrometry analysis and is of high interest. This work could significantly contribute to the knowledgebase of NMT enzymes and our understanding of NMT regulatory mechanism in promoting various protein-membrane and protein-protein interactions. Structures could help design novel high-affinity inhibitors. I find this manuscript to be appropriate for publication in Nature Communications Journal, subject to addressing my comments outlined below.

There are a number of errors related to structure refinement and results description listed below in the form of comments or questions for the authors to address. Some of the bigger issues involve: insufficient attention being paid to methods section, crystal structure refinement, lack of data that support formation of myristoylation of an N-terminal lysine side chain. More specifically, the authors need to be more careful with data presentation in the results and methods sections of the paper. Addressing these issues should improve the impact of this work.

Comments:

Title

Term "atomic resolution" usually applies to structures determined at resolution 1.2 Å or high. Authors overuse this term referring it to crystal structures determined at much lower resolutions: 2.4, 2.2 and 2.7 Å. I recommend avoiding using the "atomic resolution" term in the text and change the title of the paper. For more details see comment by A. Wlodawer & Z. Dauter (2017) 'Atomic resolution': a badly abused term in structural biology. *Acta Crys D Struct Biol* 73(4), 379-380.

Abstract

The word myristoylation of an N-terminal lysine side chain in this sentence would be a better choice as acylation is a more general term and could misinform readers.

Background

1st paragraph, 2nd sentence – Figure 1A, which shows co-translational myristoylation process of the protein, does not match correct panel on the figure.

4th paragraph, 4th sentence - Figure 1S would be more helpful if presented in the section discussing the importance of Ab-loop in the catalytic mechanism of NMT, which compares the structures of ScNMT and HsNMT.

In the same sentence, do the authors refer to Figures S1B and S1C? Do these figures correspond to serotonin N-acetyltransferase or NMT from *Saccharomyces cerevisiae* structure?

4th paragraph, last sentence - The sentence is irrelevant here and is more suitable for the results section, where the impact of B'A'-loop on the conformation of Ab-loop is discussed.

5th paragraph, 1st sentence - Do figures S1B, S1C and S1D show NMT from *Saccharomyces cerevisiae* structure?

Results

Overall structure of HsNMT1 binary and ternary complexes

The authors are not sufficiently accurate in data depiction. They discuss structures with three different peptides: X, Y, and Z. However, data statistics for structure of NMT in complex with peptide Z is not presented in Table 2. Structures of NMT with acetylated peptides V and U are not mentioned in the text. Resolution values in the text are different from the values for the complex structures shown in Table 2.

The substrate complex

Last paragraph - Could authors clarify the phrase "highly ordered with high residence time of those water molecules" in the sentence and in addition specify the alternative carboxylic side chains that bridge water molecules in HsNMT1ΔC2 structure?

The tetrahedral intermediate (TI) complex

The complex structure of HsNMT with peptide Y in intermediate (Chain A, peptide Y N-terminal amine bound to MyrCoA cofactor) and final step (Chain B, complex with products) of the reaction was determined. Careful evaluation of the density maps around ligands of the structure (PDB ID 6ehj) suggests that two alternative conformations of cofactor (MyrCoA and CoA) and peptide (tetrahedral intermediate and tetrahedral adduct) should be modeled in chain A. In addition, in the structure magnesium ion in both protein chains has poor coordination geometry and unlikely binds to phosphate groups of the cofactor. For correct assessment of metals in the protein structures, the authors should consider metal-validation server CheckMyMetal. Alternatively, anomalous difference map may also be used.

Impact of the B'A' N-terminal α-helix on product release

Two alternative NMT constructs, long (99-496) and short (110-496), were used to obtain complex structures. In all crystal structures based on the described methods, nine or ten residues at N-terminus could not be traced in the electron density maps. Thus B'A' loop (99-118 residues, Fig. S2) should not be seen in crystal structure of HsNMT:MyrCoA:X obtained with a long NMT construct.

NMT catalyses MYR on the ε-NH₂ Lys mimicking the α-NH₂ of Gly

Last paragraph - What crystal structure (with peptide V or U) is referred for structure of HsNMT1 in a complex with an acetyl-Gly-Asn starting peptide? Is this structure deposited and included into Table 3? The figure with Lys side chain bound to Myr at the active site in crystal structure should be included into the main body of the paper.

Are the catalytic parameters for HsNMT1 using MyrCoA and acetylated peptides known? Kinetic analysis for NMT in the presence of cofactor and acetylated peptide could provide additional evidences that NMT is indeed lysine myristoyltransferase. The presence of Lys-ε-Myr at the active site of NMT crystal structure could be an artifact of high peptide concentration (~ 150 mM) used in crystallization experiments.

Tables

Tables 2 and 3 - Tables with data collection and refinement statistics for all determined structures with peptides could be combined. The ligands/states and their correct names/abbreviations (X, TI-Y, MyrY, MyrZ) used in the text should be clearly stated in the Table for each NMT chain of determined complex structure.

Figures

Figure 1 - There is no need to have an independent panel showing topology of GNATs. NMT topology showing the GNAT-core fold is sufficient. It would be more informative if the authors could use a ribbon diagram of the NMT monomer with labeled secondary structure elements instead of the GNAT topology scheme presented on the figure 1A.

Figure 2 – The authors should show Fo-Fc omit map to demonstrate presence of ligands in NMT structures.

Figure 2G There is a typo - Lys9 instead of Lys8

Figure 3 Pockets allocating each amino acid of the peptide are shown with black dashed lines except pocket 5.

Methods

Methods section is missing many important details and background information.

Constructs length should be checked. It would be helpful if the authors made a separate table with all peptide sequences used in crystallization experiments. Alternatively peptide sequences could be indicated in the text. The reference siting protocol of protein expression, purification is not provided. The reference provided only for HsNMT1:MyrCoA:X structure. The details of crystallization experiments for NMT in complex with peptide Z is not mentioned. Concentrations of peptides Z, X, V and U used in crystallization experiments are not specified. The authors do not explain how the complexes of NMT with peptides Z and V were collected. In addition, information about the structure solution of the NMT structure in complex with Z, Y, and U peptides is not provided. Nothing is said about expression, purification, crystallization, data collection, structure determination and refinement for site-directed variants and NMT obtained in the presence of acetylated peptides.

Are kinetic measurements reproducible?

Point-by-point responses to reviewers' comments

Reviewer #1 (Remarks to the Author):

1) The authors represent 2Fo-Fc map of each substrate, TI and product. However, Fo-Fc map omitting with each substrate, TI and product is better to certify the model quality.

Reply: The 2Fo-Fc map of each substrate was replaced by a Fo-Fc omit map in each of the four panels (E-H) of Fig. 2 and in Fig.8E; details of the omit maps calculations are indicated in the Methods and the legend to Fig. 2 (E to H) and Fig. 8E.

2) The authors crystallized NMT1 with peptide X or peptide Y in the presence of MyrCoA. The peptide X resulted in the crystals of NMT1:MyrCoA:X (substrate) and CoA:MyrX (product) while the peptide Y resulted in the crystals of NMT1:TI-Y (intermediate) and CoA:MyrY (product). Why did the authors succeed in obtaining the intermediate complex with peptide Y, but not with peptide X? Since the intermediate state is generally thought to be unstable, the precise information how to obtain the crystal under intermediate state is useful for general readers.

Reply: Several factors have a significant impact on the success in obtaining crystals of the intermediates. These include:

- (i) soaking (peptide Y) instead of co-crystallization (peptide X) to reduce the reaction time. It is tempting to say that the 15 min soak used to solve the complex with peptide Y seems to have been quick enough to pick up the intermediate, while co-crystallization with peptide X gave time for the reaction to occur;
- (ii) both low and different pHs of the precipitant solutions used (pH 4.5 with peptide Y and pH 5.6 with peptide X), known to significantly contribute to slowing down the NMT reaction (optimum pH is 8);
- (iii) nature of the two peptides for which NMT displays different catalytic efficiencies;
- (iv) finally, many crystal trials were performed under many different conditions (i.e., different peptides, different peptide concentrations, different soaking times, etc.) and most were inconclusive (no crystal, bad resolution, no intermediate...). We only present those that were successful.

All the details of protein production and crystallization are reported in the Methods section.

3) Page7, the author described that 'we therefore solved the crystallographic structure of HsNMT1 in complex with three different peptides~ at 2.4, 2.2, and 2.7 Å resolution, respectively'. The structures complexed with peptide X and Y are new structures while the structure with peptide Z (PDB code: 5O9S described in Data availability section (page 29)) was already reported in the other paper (Castrec, B et al., 2018 Nat. Chem. Biol. 14, 671-679.). This description is very confusing. The author must cite appropriately whenever they describe known results.

Reply: This sentence has been re-written appropriately, and the Castrec paper is now cited (page 7).

4) Page8 line 4 from the bottom, 'The free N-terminal Gly2 (aa2) of peptide X is hosted in a

cavity (pocket 1, Fig. 3A) built by the side chains of Tyr180, Tyr192, Asn246, Thr282, Leu495; the C-terminal carboxylate~
Leu495 should be replaced with Gln496 because the C-terminal residues is Gln496.

Reply: Gln496 has been added. Leu495 is part of the cavity hosting the peptide. It was omitted in Fig. 3A for the sake of clarity.

5) At page 15, the author mentioned that 'further stabilizing both MyrCoA/CoA via interactions with the ribose moiety~'.

Is there any specific interaction between the B'A'-loop and the MyrCoA/CoA?

Reply: The B'A' loop is defined as the loop located between helices $\alpha B'$ and $\alpha A'$. In the ScNMT:MyrCoA structure, in which the B'A'-loop was first described (pdb 26PE), the B'A'-loop corresponds to residues 18 to 39 (21 residues). The same B'A' loop region in HsNMT1 corresponds to residues 97- 118 and is encompassed in the long NMT construct (99-496). Although this particular region folds differently in HsNMT1 compared to the B'A' loop of ScNMT (see Fig. S1) we kept the same B'A' loop name for the sake of clarity.

The B'A' loop region of HsNMT1 stabilises MyrCoA binding through different direct and indirect interactions. First, the P3 phosphate of MyrCoA/CoA makes three hydrogen bonds with the backbone nitrogens of Q118 and Y117 and the side chain of R115 (see our new Fig. 5D). Second, the B'A' loop makes interactions with the fg-loop (Fig. 5D, pink), which favour the proper planar orientation of the adenine CoA ring hydrophobic interactions provided by both the L287 and V285 side chains. The adenosine ring as a results stacks in between this pocket and the panthotenic- β -alanine side of CoA. The side chain of V286 (part of the fg-loop) is settled in a hydrophobic cavity made by the side chains of A196, M109, and A112 within the helical part of the B'A' loop region. The fg-loop is stabilised by direct hydrogen bonding between the A106 carbonyl ($\alpha B'$ helix) and the V286 backbone nitrogen. Third, the P105 residue acts as a top lid to the ribose of the adenosine moiety. Finally, the K107/E184 salt bridge strengthens the B'A'-loop fold linking the B'A'-loop with the closed conformation of the Ab-loop.

The new panel D in Fig. 5 is now reported and introduced in the new version of the manuscript and the sentence is now clarified on page 15. We have added all the above details in this part for the sake of clarity.

Q1: *Is there any specific interaction mediated by the B'A'-helix, which stabilizes the helix conformation of B'A'-helix? The interaction between Lys107 of the B'A' loop and Asp184 of Ab-loop seems to be the only interaction mediated by B'A' loop.*

Reply: Please see above for a complete answer.

Q2: *However, Lys107 is not located at B'A'-helix, but at a loop region. It is not clear to me how the B'A' Nterminal α -helix is stabilized. In addition, the residues in the B'A'-loop including Lys107 is not conserved among NMTs.*

Reply: Please, see complete answer above. In short, Fig. 5D now shows M109, A112 and R115 involved in the creation of a hydrophobic pocket. This favours direct hydrogen bonding between the A106 carbonyl ($\alpha B'$ helix) and V286 backbone nitrogen.

Q3 *Does this less conservation imply the importance of the B'A'-loop is limited to human NMT1?*

Reply: A B'A' loop can be identified in almost all NMTs (see Fig. S2A), and despite the low level of length and homology amongst the B'A' loops, it is likely that similar interaction occurs in other NMTs but whose nature is dictated by the specific NMT as suggested by the replacement of Lys107 and Glu184 by two hydrophobic residues in CaNMT. This is now reported and discussed on page 15 and 16.

6) Page 20, Fig. 13D and 13E seems to be typos (Fig S13D and S13E). The sequence of peptide U seems not to be defined. Please define the sequence (probably Ac Gly2-Asn3?).

Reply: Thanks, these typos errors have been corrected. Peptide U indeed corresponds to AcGNCFSKPR and the peptide sequences are included in the new Table 1 for clarity and in the figure legends as necessary.

7) P35, at the legend of Figure 5, interacting with Arg178 and Lys181 is typos? (Lys175 and Arg178?)

Figure 5A, "disorsered" would be "disordered".

Reply: The typos error has been corrected in the legend and in new Fig. 5.

8) Figure 2 E-H and Figure 3A-C, the peptides should be displayed in the same color. It is difficult to understand.

Reply: The color of peptides in Figs. 2 and 3 has been changed, accordingly.

9) *Figure 3, is the term 'pocket' generally used in NMT? Each pocket accommodate specific residue of the peptide for the recognition?*

Reply: The term "pocket" for NMTs was first introduced by Castrec et al. 2018 Nature Chem Biol, when the crystal structures of HsNMT1 in complex with Myr-CoA and peptide ligands defined the NMT recognition site characterized by distinct pockets, a bottleneck, and an outer salt bridge. Fig. 3A and 3B define the pockets and how each accommodate the various side chains required for peptide recognition (labeled 1-7).

10) Figure 5C, it is quite difficult to identify the labels of residues in the Ab-loop because they are too close to each other (especially D183 and D185). Please place the labels clearly to identify them easily.

Reply: New Fig. 5 is now presented; it has been modified (panel C) to take your suggestion in account. Nevertheless, it is difficult to avoid that the two residues are close to each other. Panel B shows another orientation with the two residues, completing the overall view. Finally, a new panel D shows further details involving the Ab-loop (displayed in yellow with D184 drawn as a ball and stick).

Reviewer #2 (Remarks to the Author):

Title

Term "atomic resolution" usually applies to structures determined at resolution 1.2 Å or high. Authors overuse this term referring it to crystal structures determined at much lower

resolutions: 2.4, 2.2 and 2.7 Å. I recommend avoiding using the “atomic resolution” term in the text and change the title of the paper. For more details see comment by A. Wlodawer & Z. Dauter (2017) ‘Atomic resolution’: a badly abused term in structural biology. *Acta Cryst D Struct Biol* 73(4), 379-380.

Reply: “Atomic resolution” has been replaced with “high-resolution” in the title and throughout the text.

Abstract

The word myristoylation of an N-terminal lysine side chain in this sentence would be a better choice as acylation is a more general term and could misinform readers.

Reply: Thanks, this had been adjusted.

Background

1st paragraph, 2nd sentence – Figure 1A, which shows co-translational myristoylation process of the protein, does not match correct panel on the figure.

Reply: Figure 1 was redrawn and now the co-translational myristoylation is correctly labelled in Fig. 1A.

4th paragraph, 4th sentence - Figure 1S would be more helpful if presented in the section discussing the importance of Ab-loop in the catalytic mechanism of NMT, which compares the structures of ScNMT and HsNMT.

Reply: In Fig. S1, we reported an overview of the different crystal structures of CaNMT and ScNMT, revealing distinct conformations of both B'A'- and Ab-loops. No comparison with HsNMT structures is reported in Fig S1.

The purpose of this paragraph is to clearly establish the current state-of-the-art before our findings; we believe this needs to be established very early in the text to allow the reader to easier follow the logic of the experiments. We agree that this needs a reminder in the Result section and should be associated with the more appropriate Fig. S2 which compares our HsNMT structures with that of ScNMT.

In the same sentence, do the authors refer to Figures S1B and S1C? Do these figures correspond to serotonin N-acetyltransferase or NMT from *Saccharomyces cerevisiae* structure?

Reply: The sentence refers to the Ab-loop of NMT and α_1 - α_2 loop of other GNATs such as serotonin N-acetyltransferase, shown in the topology of NMT and GNATs of Fig. 1B and 1C. This has been clarified in the sentence, and a new Fig. 1 has been provided accordingly also to reviewer 1 comments.

4th paragraph, last sentence - The sentence is irrelevant here and is more suitable for the *results section*, where the impact of B'A'-loop on the conformation of Ab-loop is discussed.

Reply: The sentence, as stated above, is important to establish the context.

5th paragraph, 1st sentence - Do figures S1B, S1C and S1D show NMT from *Saccharomyces cerevisiae* structure?

Reply: Yes, they do. Fig. S1B, S1C, and S1D reported different crystal structures of ScNMT and only Fig. S1D shows the ScNMT structure with a non-hydrolysable MyrCoA analogue (S-(2-oxo)-pentadecyl-CoA, NHM), as reported in the text.

Results

Overall structure of HsNMT1 binary and ternary complexes

The authors are not sufficiently accurate in data depiction. They discuss structures with three different peptides: X, Y, and Z. However, data statistics for structure of NMT in complex with peptide Z is not presented in Table 2. Structures of NMT with acetylated peptides V and U are not mentioned in the text. Resolution values in the text are different from the values for the complex structures shown in Table 2.

Reply: We recently reported the structure of HsNMT with peptide Z (Castrec et al. 2018 Nature Chem Biol.). The sentence concerning the description of the three structures X, Y and Z has been re-written to clarify this point taking in account also reviewer 1's suggestion.

The description of the structures of HsNMT1 with peptides U and V is mentioned in the paragraph entitled "NMT catalyses MYR on the ϵ -NH₂ of Lys mimicking the α -NH₂ of Gly", describing new Fig. 8C (peptide U) and Fig. 8D and 8E (peptide V). We have rewritten the legend of Fig. 8 to be clearer and the sequences of peptides X, Y, U and V have been added to the new Table 1. Taking your comment in account, Fig. 8 is now added to the main body of the Figures and corresponds to previous Fig. S13. Resolution values written in the text have been corrected to be the same as reported in Table 1.

The substrate complex

Last paragraph - Could authors clarify *the phrase "highly ordered with high residence time of those water molecules" in the sentence and in addition specify the alternative carboxylic side chains that bridge water molecules in HsNMT1 Δ C2 structure?*

Reply: The sentence has been reformulated for clarity.

The alternative carboxylic side chains that bridges water molecules in the HsNMT1 Δ C2 structure is Glu244, which interacts with Gly2 via two molecules of the water channel. This is explained in the same paragraph earlier when describing Fig. S4.

The tetrahedral intermediate (TI) complex

The complex structure of HsNMT with peptide Y in intermediate (Chain A, peptide Y N-terminal amine bound to MyrCoA cofactor) and final step (Chain B, complex with products) of the reaction was determined. Careful evaluation of the density maps around ligands of the structure (PDB ID 6ehj) suggests that two alternative conformations of cofactor (MyrCoA and CoA) and peptide (tetrahedral intermediate and tetrahedral adduct) should be modeled in chain A.

Reply: We have re-refined the structure with the intermediate and products of the reaction bound at chain A at different occupancies. Our results show that we indeed have a mixture of both species, with best refinement result at 0.3 occupancy for the intermediate and 0.7

occupancy of the products of the reaction. This was clear from the Fo-Fc map, which initially showed a peak of negative electron density that faded almost completely after refining both species at the occupancies mentioned. This negative peak of electron density observed in the Fo-Fc map was believed to be the result of noise and/or radiation damage, as for other peaks observed in the map which cannot be justified in another way. We very much appreciate to have been pointed at this as well as the resulting improvement in the model.

In addition, in the structure magnesium ion in both protein chains has poor coordination geometry and unlikely binds to phosphate groups of the cofactor. For correct assessment of metals in the protein structures, the authors should consider metal-validation server CheckMyMetal. Alternatively, anomalous difference map may also be used.

Reply: After inspection, the electron density for these atoms is not particularly intense and thus we ran a refinement substituting them by water molecules. Waters refined well and thus magnesium atoms have been substituted and refined as water molecules in the final model.

Impact of the B'A' N-terminal α -helix on product release

Two alternative NMT constructs, long (99-496) and short (110-496), were used to obtain complex structures. In all crystal structures based on the described methods, nine or ten residues at N-terminus could not be traced in the electron density maps. Thus B'A' loop (99-118 residues, Fig. S2) should not be seen in crystal structure of HsNMT:MyrCoA:X obtained with a long NMT construct.

Reply: We apologize for the lack of clarity that could lead to a misunderstanding. As reported previously, in the ScNMT:MyrCoA structure, the B'A'-loop corresponds to residues 18 to 39 (21 residues). The same B'A'-loop region in HsNMT1 corresponds to residues 97-118 and is present only in the long NMT construct (99-496). This particular region folds differently in HsNMT1 to the B'A' loop of ScNMT (see Figure S1), but we kept the same B'A' loop name for clarity. We could trace 70% of this region in HsNMT1, only the first 6 N-terminal residues (residues 99-104) are found to be disordered. The B'A' loop region is observed in structures in complex with peptides X, U, V and S, obtained using the long NMT construct, with the exception of Z peptide, in which CoA released induced disordered of the B'A'-loop region. Therefore the B'A' loop region is observed in the long NMT construct. For clarity, we have modified both the sentence and the numbers on page 27.

NMT catalyses MYR on the ϵ -NH₂ Lys mimicking the α -NH₂ of Gly

Last paragraph - What crystal structure (with peptide V or U) is referred for structure of HsNMT1 in a complex with an acetyl-Gly-Asn starting peptide? Is this structure deposited and included into Table 3? The figure with Lys side chain bound to Myr at the active site in crystal structure should be included into the main body of the paper.

Reply: A new Table 1 now provides data related to all wild-type HsNMT1 crystal structures, including those with peptide V and U (formerly in Table 3). These structures are deposited in the PDB under codes 6SJZ (peptide U), and 6SK2 (peptide V). They will be released once the manuscript is published. Figure S13 with the Lys side chain bound to Myr in the crystal structure is now included in the main body of the paper as suggested (new Fig. 8 with new panels D/E).

Are the catalytic parameters for HsNMT1 using MyrCoA and acetylated peptides known? Kinetic analysis for NMT in the presence of cofactor and acetylated peptide could provide

additional evidences that NMT is indeed lysine myristoyltransferase. The presence of Lys- ϵ -Myr at the active site of NMT crystal structure could be an artifact of high peptide concentration (~ 150 mM) used in crystallization experiments.

Reply: The co-crystallization conditions for Lys Myr crystals are detailed on page 26. In this case, the HsNMT1:MyrCoA:peptide complex (ratio 1:1.5:1.5) is 7.5 mg/mL. This corresponds to 160 μ M HsNMT and 250 μ M peptide. This is now indicated. This is not such a high peptide concentration as K_m values are usually in the 10-2,000 μ M range (see also Table 2) and we do use ranges up to 2 mM for in vitro kinetics. We have already performed kinetic analysis with specific Lys MYRed substrates. These and other data with many more peptides will be the topic of another article, which we are now completing. Of note, we have already reported that N-terminal substrates have a high dynamic range of k_{cat}/K_m values (5-50,000 $M^{-1} s^{-1}$). In this case, the data we have in hands show values in the range of 250 $M^{-1} s^{-1}$, which is fully relevant in vivo and confirms that this reaction is not an artefact. (in vitro catalysis is usually $<0.01 M^{-1} s^{-1}$).

We would also like to clarify that 150 mM of peptide was only used in soaking experiments with peptide Y.

Tables

Tables 2 and 3 - Tables with data collection and refinement statistics for all determined structures with peptides could be combined.

Reply: Tables with data collection and refinement statistics with peptide X, Y, U, and V are now fused into a single Table and Table 1. Data collection and refinement statistics concerning structures with peptides S and T are reported in Table S1.

The ligands/states and their correct names/abbreviations (X, T1-Y, MyrY, MyrZ) used in the text should be clearly stated in the Table for each NMT chain of determined complex structure.

Reply: This is now indicated in Table 1, line 3 and 4.

Figures

Figure 1 - There is no need to have an independent panel showing topology of GNATs. NMT topology showing the GNAT-core fold is sufficient. It would be more informative if the authors could use a ribbon diagram of the NMT monomer with labeled secondary structure elements instead of the GNAT topology scheme presented on the figure 1A.

Reply: A new complete Fig. 1 has been provided accordingly.

Figure 2 – The authors should show Fo-Fc omit map to demonstrate presence of ligands in NMT structures.

Reply: New Fig. 2E-H and Fig.8E panels are now provided where the former 2Fo-Fc maps were replaced by Fo-Fc omit maps.

Figure 2G There is a typo - Lys9 instead of Lys8

Reply: The typo error has been corrected.

Figure 3 Pockets allocating each amino acid of the peptide are shown with black dashed lines except pocket 5.

Reply: Pocket 5 of Fig. 3A is now shown with black dashed.

Methods

Methods section is missing many important details and background information.

Constructs length should be checked. It would be helpful if the authors made a separate table with all peptide sequences used in crystallization experiments.

Alternatively peptide sequences could be indicated in the text. The reference siting protocol of protein expression, purification is not provided.

The reference provided only for HsNMT1:MyrCoA:X structure.

Reply: Construct lengths were double checked and the misunderstood points clarified (see also answers relating to the B'A' N-terminal α -helix).

The sequences of all peptides used in crystallization experiments have been added to Table 1 and Table S1 and, when necessary, recalled it in the appropriate Figure legend. This will help the reader to immediately visualize the sequence through the text or in the Figures.

References for protein expression and purification are provided in the Methods (Goncalves et al. Anal Biochem 2012 for the short NMT and Castrec et al. Nature Chem Biol for the long HsNMT).

The details of crystallization experiments for NMT in complex with peptide Z is not mentioned.

Reply: As now clarified in the Results, the crystallization of HsNMT1 in complex with peptide Z was previously reported and further analysed here in light of the new X and Y complexes.

Concentrations of peptides Z, X, V and U used in crystallization experiments are not specified.

Reply: Ratios and concentrations are now reported in the Methods.

The authors do not explain how the complexes of NMT with peptides Z and V were collected.

In addition, information about the structure solution of the NMT structure in complex with Z, Y, and U peptides is not provided. Nothing is said about expression, purification,

Reply: Expression and purification of long HsNMT was done as previously reported and a reference is provided. Information about the structure solutions of the NMT in complex with Y, U and U is reported in the Methods, whereas that of Z reference is cited.

Crystallization, data collection, structure determination and refinement for site-directed variants and NMT obtained in the presence of acetylated peptides.

Reply: This is reported in the Supplementary Information.

Are kinetic measurements reproducible?

Reply: Reproducibility on the kinetic measurements was assessed through purification of various batches of the variant and repeated kinetics. The resulting standard deviation is provided in Table 2.

REVIEWERS' COMMENTS:

Reviewer #1 (Remarks to the Author):

The authors have appropriately responded to the previous concerns.

Reviewer #2 (Remarks to the Author):

The manuscript entitled "High-resolution snapshots of human N-myristoyltransferase in action illuminate a unique mechanism promoting both N-terminal Lys and Gly myristoylation" has been carefully revised by Dr Giglione and co-workers. Authors addressed my comments concerning the interpretation of results, structure refinement of the HsNMT complex with peptide Y, edited materials and method section including important information about protein production, crystallization, and structure determination. The information about constructs and peptide sequences is now clearly indicated in the text, tables with structure data statistics and figures were revised and provided in accordance with raised questions. I believe that revised work resulted in an improved manuscript. I am satisfied with authors responses to my comments and questions. I have only one comment for the authors to address listed below.

Comment for the authors:

The re-refined structure of HsNMT in complex with peptide Y showing alternative conformation of the cofactor and peptide at the protein active site should be redeposited and PDB ID of the structure updated in revised version of the manuscript.

Point-by-point response to any issues raised by our referees

REVIEWERS' COMMENTS:

Reviewer #1 (Remarks to the Author):

The authors have appropriately responded to the previous concerns.

Reviewer #2 (Remarks to the Author):

The manuscript entitled "High-resolution snapshots of human N-myristoyltransferase in action illuminate a unique mechanism promoting both N-terminal Lys and Gly myristoylation" has been carefully revised by Dr Giglione and co-workers. Authors addressed my comments concerning the interpretation of results, structure refinement of the HsNMT complex with peptide Y, edited materials and method section including important information about protein production, crystallization, and structure determination. The information about constructs and peptide sequences is now clearly indicated in the text, tables with structure data statistics and figures were revised and provided in accordance with raised questions. I believe that revised work resulted in an improved manuscript. I am satisfied with authors responses to my comments and questions. I have only one comment for the authors to address listed below.

Comment for the authors: The re-refined structure of HsNMT in complex with peptide Y showing alternative conformation of the cofactor and peptide at the protein active site should be redeposited and PDB ID of the structure updated in revised version of the manuscript.

We have updated the existing entry with the new coordinates; we do not need to change the pdb code, which remains the same.